# Transformers Learn to Achieve Second-Order Convergence Rates for In-Context Linear Regression

**Deqing Fu**     **Tian-Qi Chen**     **Robin Jia**     **Vatsal Sharan**
Department of Computer Science
University of Southern California
`{deqingfu,tchen939,robinjia,vsharan}@usc.edu`

## Abstract

Transformers excel at *in-context learning* (ICL)—learning from demonstrations without parameter updates—but how they do so remains a mystery. Recent work suggests that Transformers may internally run Gradient Descent (GD), a first-order optimization method, to perform ICL. In this paper, we instead demonstrate that Transformers learn to approximate second-order optimization methods for ICL. For in-context linear regression, Transformers share a similar convergence rate as *Iterative Newton's Method*; both are exponentially faster than GD. Empirically, predictions from successive Transformer layers closely match different iterations of Newton's Method *linearly*, with each middle layer roughly computing 3 iterations; thus, Transformers and Newton's method converge at roughly the same rate. In contrast, Gradient Descent converges *exponentially* more slowly. We also show that Transformers can learn in-context on ill-conditioned data, a setting where Gradient Descent struggles but Iterative Newton succeeds. Finally, to corroborate our empirical findings, we prove that Transformers can implement $k$ iterations of Newton's method with $k + \mathcal{O}(1)$ layers.

## 1   Introduction

Transformer neural networks [Vaswani et al., 2017] have become the default architecture for natural language processing [Devlin et al., 2019, Brown et al., 2020, OpenAI, 2023]. As first demonstrated by GPT-3 [Brown et al., 2020], Transformers excel at *in-context learning* (ICL)—learning from prompts consisting of input-output pairs, without updating model parameters. Through in-context learning, Transformer-based Large Language Models (LLMs) can achieve state-of-the-art few-shot performance across a variety of downstream tasks [Rae et al., 2022, Smith et al., 2022, Thoppilan et al., 2022, Chowdhery et al., 2022].

Given the importance of Transformers and ICL, many prior efforts have attempted to understand how Transformers perform in-context learning. Prior work suggests Transformers can approximate various linear functions well in-context [Garg et al., 2022]. Specifically to linear regression tasks, prior work has tried to understand the ICL mechanism, and the dominant hypothesis is that Transformers learn in-context by running optimization internally through gradient-based algorithms [von Oswald et al., 2022, 2023, Ahn et al., 2023, Dai et al., 2023, Mahankali et al., 2024].

This paper presents strong evidence for a competing hypothesis: Transformers trained to perform in-context linear regression learn a strategy much more similar to a second-order optimization method than a first-order method like Gradient Descent (GD). In particular, Transformers approximately implement a second-order method with a convergence rate very similar to Newton-Schulz's Method, also known as the *Iterative Newton's Method*, which iteratively improves an estimate of the inverse of

---

Our codes are available at `https://github.com/DeqingFu/transformers-icl-second-order`.

38th Conference on Neural Information Processing Systems (NeurIPS 2024).

the data matrix to compute the optimal weight vector. Across many Transformer layers, subsequent layers approximately compute more and more iterations of Newton's Method, with increasingly better predictions; both eventually converge to the optimal minimum-norm solution found by ordinary least squares (OLS). Interestingly, this mechanism is specific to Transformers: LSTMs do not learn these same second-order methods, as their predictions do not even improve across layers.

We present both empirical and theoretical evidence for our claims. Empirically, Transformer layers demonstrate a similar rate of convergence to the OLS solution as second-order methods such as Iterative Newton, which is substantially faster than the rate of convergence of GD (Figure 2). The predictions made by the Transformer at successive layers closely match the predictions made by Iterative Newton after a proportional number of iterations, showing that they progress in similar ways at the same rate. In contrast, to match the Transformer's predictions after $k$ layers, GD would have to run for exponential in $k$ many steps (Figure 3). Some individual Transformer layers make progress equivalent to hundreds of GD steps: these layers must be doing something more sophisticated than GD. Furthermore, a crucial aspect of second-order methods is that they can handle ill-conditioned problems by correcting the curvature. We find that the convergence rate of Transformers is not significantly affected by ill-conditioning, which again matches Iterative Newton but not GD. To provide theoretical grounding to our empirical results, we show that Transformer circuits can efficiently implement Iterative Newton: one transformer layer can compute one Newton iteration (given $\mathcal{O}(1)$ pre/post-processing layers), and requires hidden states of dimension $\mathcal{O}(d)$ for a $d$-dimensional linear regression problem. Overall, our work provides a mechanistic account of how Transformers perform ICL that explains model behavior better than previous hypotheses, and hints at why Transformers are well-suited for ICL relative to other architectures.

## 2 Related Work

**In-context learning by large language models.** GPT-3 [Brown et al., 2020] first showed that Transformer-based large language models can "learn" to perform new tasks from in-context demonstrations (i.e., input-output pairs). Since then, a large body of work in NLP has studied in-context learning, for instance by understanding how the choice and order of demonstrations affects results [Lu et al., 2022, Liu et al., 2022, Rubin et al., 2022, Su et al., 2023, Chang and Jia, 2023, Nguyen and Wong, 2023], studying the effect of label noise [Min et al., 2022c, Yoo et al., 2022, Wei et al., 2023], and proposing methods to improve ICL accuracy [Zhao et al., 2021, Min et al., 2022a,b].

**In-context learning beyond natural language.** Inspired by the phenomenon of ICL by large language models, subsequent work has studied how Transformers learn in-context beyond NLP tasks. Garg et al. [2022] first investigated Transformers' ICL abilities for various classical machine learning problems, including linear regression. We largely adopt their linear regression setup in this work. Li et al. [2023] formalize in-context learning as an algorithm learning problem. Han et al. [2023] suggests that Transformers learn in-context by performing Bayesian inference on prompts, which can be asymptotically interpreted as kernel regression. Other work has analyzed how Transformers do in-context classification [Tarzanagh et al., 2023a,b, Zhang et al., 2023], the role of pertaining data [Raventós et al., 2023], and the relationship between model architecture and ICL [Lee et al., 2023].

**Do Transformers implement Gradient Descent?** A growing body of work has suggested that Transformers learn in-context by implementing gradient descent within their internal representations. Akyürek et al. [2022] summarize operations that Transformers can implement, such as multiplication and affine transformations, and show that Transformers can implement gradient descent for linear regression using these operations. Concurrently, von Oswald et al. [2022] argue that Transformers learn in-context via gradient descent, where one layer performs one gradient update. In subsequent work, von Oswald et al. [2023] further argue that Transformers are strongly biased towards learning to implement gradient-based optimization routines. Ahn et al. [2023] extend the work of von Oswald et al. [2022] by showing Transformers can learn to implement preconditioned Gradient Descent, where the pre-conditioner can adapt to the data. Bai et al. [2023] provide detailed constructions for how Transformers can implement a range of learning algorithms via gradient descent. Finally, Dai et al. [2023] conduct experiments on NLP tasks and conclude that Transformer-based language models performing ICL behave similarly to models fine-tuned via gradient descent; however, concurrent work [Shen et al., 2023b] argues that real-world LLMs do not perform ICL via gradient descent. Mahankali et al. [2024] showed that implementing gradient descent is a global minima for single layer linear self-attention. However, we study deeper models in this work, which can behave differently from

single-layer models. In this paper, we argue that Transformers actually learn to perform in-context learning by implementing a second-order optimization method, not gradient descent[1].

**Mechanistic interpretability for Transformers.** Our work attempts to understand the mechanism through which Transformers perform in-context learning. Prior work has studied other aspects of Transformers' internal mechanisms, including reverse-engineering language models [Wang et al., 2022], the grokking phenomenon [Power et al., 2022, Nanda et al., 2023], manipulating attention maps [Hassid et al., 2022], and circuit finding [Conmy et al., 2023].

**Theoretical Expressivity of Transformers.** Giannou et al. [2023] provide a construction of looped transformers to implement Iterative Newton's method for solving pseudo-inverse, and each Newton iteration can be implemented by 13 looped Transformer layers. In contrast, our construction needs only one Transformer layer to compute one Newton iteration.

## 3 Problem Setup

In this paper, we focus on the following linear regression task. The task involves $n$ examples $\{\boldsymbol{x}_i, y_i\}_{i=1}^n$ where $\boldsymbol{x}_i \in \mathbb{R}^d$ and $y_i \in \mathbb{R}$. The examples are generated from the following data generating distribution $P_{\mathcal{D}}$, parameterized by a distribution $\mathcal{D}$ over $(d \times d)$ positive semi-definite matrices. For each sequence of $n$ in-context examples, we first sample a ground-truth weight vector $\boldsymbol{w}^\star \overset{\text{i.i.d.}}{\sim} \mathcal{N}(\boldsymbol{0}, \boldsymbol{I}) \in \mathbb{R}^d$ and a matrix $\boldsymbol{\Sigma} \overset{\text{i.i.d.}}{\sim} \mathcal{D}$. For $i \in [n]$, we sample each $\boldsymbol{x}_i \overset{\text{i.i.d.}}{\sim} \mathcal{N}(\boldsymbol{0}, \boldsymbol{\Sigma})$. The label $y_i$ for each $\boldsymbol{x}_i$ is given by $y_i = \boldsymbol{w}^{\star\top} \boldsymbol{x}_i$. Note that for much of our experiments

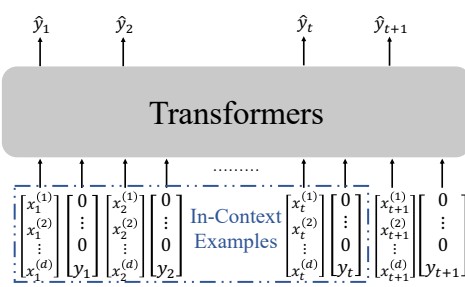

Figure 1: Illustration of how Transformers are trained to do in-context linear regression.

$\mathcal{D}$ is only supported on the identity matrix $\boldsymbol{I}$ and hence $\boldsymbol{\Sigma} = \boldsymbol{I}$, but we also consider some distributions over ill-conditioned matrices, which give rise to ill-conditioned regression problems. Most of our results are on this noiseless setup and results with the noisy setup are in Appendix A.3.2.

### 3.1 Standard Methods for Solving Linear Regression

Our central research question is:

> ***What convergence rate does the algorithm Transformers learn for linear regression achieve?***

To investigate this question, we first discuss various known algorithms for linear regression. We then compare them with Transformers empirically in §4 and theoretically in §5, to evaluate if Transformers are more similar to first-order or second-order methods. We care particularly about algorithms' convergence rates (the number of steps required to reach an $\epsilon$ error).

For any time step $t$, let $\boldsymbol{X}^{(t)} = [\boldsymbol{x}_1 \ \cdots \ \boldsymbol{x}_t]^\top$ be the data matrix and $\boldsymbol{y}^{(t)} = [y_1 \ \cdots \ y_t]^\top$ be the labels for all the datapoints seen so far. Note that since $t$ can be smaller than the data dimension $d$, $\boldsymbol{X}^{(t)}$ can be singular. We now consider various algorithms for making predictions for $\boldsymbol{x}_{t+1}$ based on $\boldsymbol{X}^{(t)}$ and $\boldsymbol{y}^{(t)}$. When it is clear from context, we drop the superscript and refer to $\boldsymbol{X}^{(t)}$ and $\boldsymbol{y}^{(t)}$ as $\boldsymbol{X}$ and $\boldsymbol{y}$, where $\boldsymbol{X}$ and $\boldsymbol{y}$ correspond to all the datapoints seen so far.

**Ordinary Least Squares.** This method finds the minimum-norm solution to the objective:

$$\mathcal{L}(\boldsymbol{w} \mid \boldsymbol{X}, \boldsymbol{y}) = \frac{1}{2n} \|\boldsymbol{y} - \boldsymbol{X}\boldsymbol{w}\|_2^2. \tag{1}$$

The Ordinary Least Squares (OLS) solution has a closed form given by the Normal Equations:

$$\hat{\boldsymbol{w}}^{\text{OLS}} = (\boldsymbol{X}^\top \boldsymbol{X})^\dagger \boldsymbol{X}^\top \boldsymbol{y} \tag{2}$$

---

[1]After an initial version of this paper, Vladymyrov et al. [2024] found that a variant of Gradient Descent can mimic Iterative Newton by approximating the inverse implicitly and getting second-order rates, which also supports our claim.

where $\boldsymbol{S} := \boldsymbol{X}^\top \boldsymbol{X}$ and $\boldsymbol{S}^\dagger$ is the pseudo-inverse [Moore, 1920] of $\boldsymbol{S}$.

**Gradient Descent.** Gradient descent (GD) is a first-order method which finds the weight vector $\hat{\boldsymbol{w}}^{\mathrm{GD}}$ with initialization $\hat{\boldsymbol{w}}_0^{\mathrm{GD}} = \boldsymbol{0}$ using the iterative update rule:

$$\hat{\boldsymbol{w}}_{k+1}^{\mathrm{GD}} = \hat{\boldsymbol{w}}_k^{\mathrm{GD}} - \eta \nabla_{\boldsymbol{w}} \mathcal{L}(\hat{\boldsymbol{w}}_k^{\mathrm{GD}} \mid \boldsymbol{X}, \boldsymbol{y}). \tag{3}$$

It is known that GD requires $\mathcal{O}\left(\kappa(\boldsymbol{S}) \log(1/\epsilon)\right)$ steps to converge to an $\epsilon$ error where $\kappa(\boldsymbol{S}) = \frac{\lambda_{\max}(\boldsymbol{S})}{\lambda_{\min}(\boldsymbol{S})}$ is the *condition number*. Thus, when $\kappa(\boldsymbol{S})$ is large, GD converges slowly [Boyd and Vandenberghe, 2004].

**Online Gradient Descent.** While GD computes the gradient with respect to the full data matrix $\boldsymbol{X}$ at each iteration, Online Gradient Descent (OGD) is an online algorithm that only computes gradients on the newly received data point $\{\boldsymbol{x}_k, y_k\}$ at step $k$:

$$\hat{\boldsymbol{w}}_{k+1}^{\mathrm{OGD}} = \hat{\boldsymbol{w}}_k^{\mathrm{OGD}} - \eta_k \nabla_{\boldsymbol{w}} \mathcal{L}(\hat{\boldsymbol{w}}_k^{\mathrm{OGD}} \mid \boldsymbol{x}_k, y_k). \tag{4}$$

Picking $\eta_k = \frac{1}{\|\boldsymbol{x}_k\|_2^2}$ ensures that the new weight vector $\hat{\boldsymbol{w}}_{k+1}^{\mathrm{OGD}}$ makes zero error on $\{\boldsymbol{x}_k, y_k\}$.

**Iterative Newton's Method.** This is a second-order method which finds the weight vector $\hat{\boldsymbol{w}}^{\mathrm{Newton}}$ by iteratively apply Newton's method to finding the pseudo inverse of $\boldsymbol{S} = \boldsymbol{X}^\top \boldsymbol{X}$ [Schulz, 1933, Ben-Israel, 1965].

$$\boldsymbol{M}_0 = \alpha \boldsymbol{S}, \text{ where } \alpha = \frac{2}{\|\boldsymbol{S}\boldsymbol{S}^\top\|_2}, \quad \hat{\boldsymbol{w}}_0^{\mathrm{Newton}} = \boldsymbol{M}_0 \boldsymbol{X}^\top \boldsymbol{y},$$
$$\boldsymbol{M}_{k+1} = 2\boldsymbol{M}_k - \boldsymbol{M}_k \boldsymbol{S} \boldsymbol{M}_k, \quad \hat{\boldsymbol{w}}_{k+1}^{\mathrm{Newton}} = \boldsymbol{M}_{k+1} \boldsymbol{X}^\top \boldsymbol{y}. \tag{5}$$

This computes an approximation of the psuedo inverse using the moments of $\boldsymbol{S}$. In contrast to GD, the Iterative Newton's method only requires $\mathcal{O}(\log \kappa(\boldsymbol{S}) + \log \log(1/\epsilon))$ steps to converge to an $\epsilon$ error [Soderstrom and Stewart, 1974, Pan and Schreiber, 1991]. Note that this is exponentially faster than the convergence rate of GD. We discuss additional algorithms such as Conjugate Gradient, BFGS, and L-BFGS in the Appendix A.2.3.

## 3.2 Solving Linear Regression with Transformers

We will use neural network models such as Transformers to solve this linear regression task. As shown in Figure 1, at time step $t + 1$, the model sees the first $t$ in-context examples $\{\boldsymbol{x}_i, y_i\}_{i=1}^t$, and then makes predictions for $\boldsymbol{x}_{t+1}$, whose label $y_{t+1}$ is not observed by the Transformers model.

We randomly initialize our models and then train them on the linear regression task to make predictions for every number of in-context examples $t$, where $t \in [n]$. Training and test data are both drawn from $P_{\mathcal{D}}$. To make the input prompts contain both $\boldsymbol{x}_i$ and $y_i$, we follow same the setup as Garg et al. [2022]'s to zero-pad $y_i$'s, and use the same GPT-2 model [Radford et al., 2019] with softmax activation and causal attention mask (discussed later in Definition 3.1).

We now present the key mathematical details for the Transformer architecture, and how they can be used for in-context learning. First, the causal attention mask enforces that attention heads can only attend to hidden states of previous time steps, and is defined as follows.

**Definition 3.1** (Causal Attention Layer). A **causal** attention layer with $M$ heads and activation function $\sigma$ is denoted as $\mathrm{Attn}$ on any input sequence $\boldsymbol{H} = [\boldsymbol{h}_1, \cdots, \boldsymbol{h}_N] \in \mathbb{R}^{D \times N}$, where $D$ is the dimension of hidden states and $N$ is the sequence length. In the vector form,

$$\tilde{\boldsymbol{h}}_t = [\mathrm{Attn}(\boldsymbol{H})]_t = \boldsymbol{h}_t + \sum_{m=1}^{M} \sum_{j=1}^{t} \sigma\left(\langle \boldsymbol{Q}_m \boldsymbol{h}_t, \boldsymbol{K}_m \boldsymbol{h}_j \rangle\right) \cdot \boldsymbol{V}_m \boldsymbol{h}_j. \tag{6}$$

Vaswani et al. [2017] originally proposed the Transformer architecture with the Softmax activation function for the attention layers. Later works have found that replacing $\mathrm{Softmax}(\cdot)$ with $\frac{1}{t}\mathrm{ReLU}(\cdot)$ does not hurt model performance [Cai et al., 2022, Shen et al., 2023a, Wortsman et al., 2023]. The Transformers architecture is defined by putting together attention layers with feed forward layers:

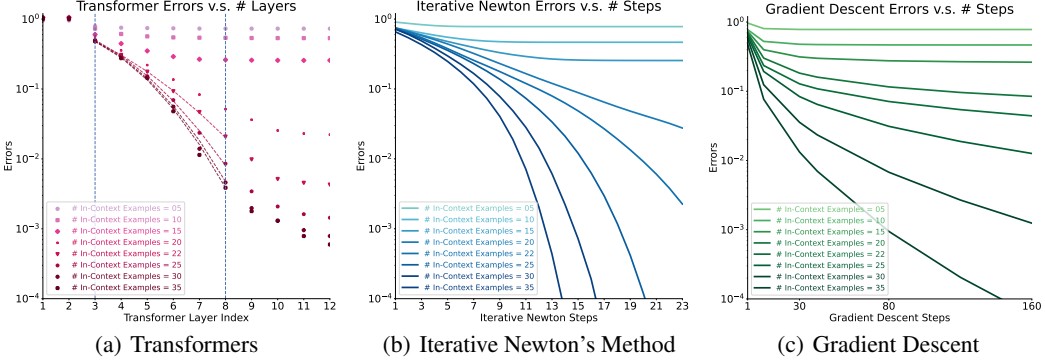

(a) Transformers     (b) Iterative Newton's Method     (c) Gradient Descent

Figure 2: **Convergence of Algorithms.** Similar to Iterative Newton and GD, Transformer's performance improve over the layer index $\ell$. When $n > d$, the Transformer model, from layers 3 to 8, demonstrates a superlinear convergence rate, similar to Iterative Newton, while GD, with fixed step size, is sublinear. Later layers of Transformers show a slower convergence rate, and we hypothesize they have little incentive to implement the algorithm precisely since the error is already very small. A 24-layer Transformer model exhibits the same superlinear convergence (Figure 25 in §A.4.2).

**Definition 3.2** (Transformers). An $L$-layer decoder-based transformer with Causal Attention Layers is denoted as $\text{TF}_{\boldsymbol{\theta}}$ and is a composition of a MLP Layer (with a skip connection) and a Causal Attention Layers. For input sequence $\boldsymbol{H}^{(0)}$, the transformers $\ell$-th hidden layer is given by

$$\text{TF}_{\boldsymbol{\theta}}^{\ell}(\boldsymbol{H}^{(0)}) := \boldsymbol{H}^{(\ell)} = \text{MLP}_{\boldsymbol{\theta}_{\text{mlp}}^{(\ell)}}\left(\text{Attn}_{\boldsymbol{\theta}_{\text{attn}}^{(\ell)}}(\boldsymbol{H}^{(\ell-1)})\right).$$

where $\boldsymbol{\theta} = \{\boldsymbol{\theta}_{\text{mlp}}^{(\ell)}, \boldsymbol{\theta}_{\text{attn}}^{(\ell)}\}_{\ell=1}^{L}$ and $\boldsymbol{\theta}_{\text{attn}}^{(\ell)} = \{\boldsymbol{Q}_m^{(\ell)}, \boldsymbol{K}_m^{(\ell)}, \boldsymbol{V}_m^{(\ell)}\}_{m=1}^{M}$ has $M$ heads at layer $\ell$.

In particular for the linear regression task, Transformers perform in-context learning as follows

**Definition 3.3** (Transformers for Linear Regression). Given in-context examples $\{\boldsymbol{x}_1, y_1, \ldots, \boldsymbol{x}_t, y_t\}$, Transformers make predictions on a query example $\boldsymbol{x}_{t+1}$ through a readout layer parameterized as $\boldsymbol{\theta}_{\text{readout}} = \{\boldsymbol{u}, v\}$, and the prediction $\hat{y}_{t+1}^{\text{TF}}$ is given by

$$\hat{y}_{t+1}^{\text{TF}} := \text{ReadOut}\Big[\underbrace{\text{TF}_{\boldsymbol{\theta}}^{L}(\{\boldsymbol{x}_1, \boldsymbol{y}_1, \cdots, \boldsymbol{x}_t, \boldsymbol{y}_t, \boldsymbol{x}_{t+1}\})}_{\boldsymbol{H}^{(L)}}\Big] = \boldsymbol{u}^{\top}\boldsymbol{H}_{:,2t+1}^{(L)} + v.$$

To compare the rate of convergence of iterative algorithms to that of Transformers, we treat the layer index $\ell$ of Transformers as analogous to the iterative step $k$ of algorithms discussed in §3.1. Note that for Transformers, we need to re-train the $\text{ReadOut}$ layer for every layer index $\ell$ so that they can improve progressively (see §4.1 and for experimental details) for linear regression tasks.

### 3.3 Measuring Algorithmic Similarity

We propose two metrics to measure the similarity between linear regression algorithms.

**Similarity of Errors.** This metric aims to measure similarity of algorithms through comparing prediction errors. For a linear regression algorithm $\mathcal{A}$, let $\mathcal{A}(\boldsymbol{x}_{t+1} \mid \{\boldsymbol{x}_i, y_i\}_{i=1}^{t})$ denote its prediction on the $(t+1)$-th in-context example $\boldsymbol{x}_{t+1}$ after observing the first $t$ examples (see Figure 1). We write $\mathcal{A}(\boldsymbol{x}_{t+1}) := \mathcal{A}(\boldsymbol{x}_{t+1} \mid \{\boldsymbol{x}_i, y_i\}_{i=1}^{t})$ for brevity. Errors (i.e., residuals) on the sequence are:[2]

$$\mathcal{E}(\mathcal{A} \mid \{\boldsymbol{x}_i, y_i\}_{i=1}^{n+1}) = \Big[\mathcal{A}(\boldsymbol{x}_2) - y_2, \cdots, \mathcal{A}(\boldsymbol{x}_{n+1}) - y_{n+1}\Big]^{\top}.$$

The similarity of errors for two algorithms $\mathcal{A}_a$ and $\mathcal{A}_b$ is the expected cosine similarity of their errors on a randomly sampled data sequence:

$$\text{SimE}(\mathcal{A}_a, \mathcal{A}_b) = \mathop{\mathbb{E}}_{\{\boldsymbol{x}_i, y_i\}_{i=1}^{n+1} \sim P_{\mathcal{D}}}\left[\mathcal{C}\Big(\mathcal{E}(\mathcal{A}_a|\{\boldsymbol{x}_i, y_i\}_{i=1}^{n+1}), \mathcal{E}(\mathcal{A}_b|\{\boldsymbol{x}_i, y_i\}_{i=1}^{n+1})\Big)\right].$$

---

[2]the indices start from 2 to $n+1$ because we evaluate all cases where $t$ can choose from $1, \cdots, n$.

Here $\mathcal{C}(\boldsymbol{u}, \boldsymbol{v}) = \frac{\langle \boldsymbol{u}, \boldsymbol{v} \rangle}{\|\boldsymbol{u}\|_2 \|\boldsymbol{v}\|_2}$ is the cosine similarity, $n$ is the total number of in-context examples, and $P_{\mathcal{D}}$ is the data generation process discussed previously.

**Similarity of Induced Weights.** All standard algorithms for linear regression estimate a weight vector $\hat{\boldsymbol{w}}$. While neural ICL models like Transformers do not explicitly learn such a weight vector, similar to Akyürek et al. [2022], we can *induce* an implicit weight vector $\tilde{\boldsymbol{w}}$ learned by any algorithm $\mathcal{A}$ by fitting a weight vector to its predictions. We can then measure similarity of algorithms by comparing the induced $\tilde{\boldsymbol{w}}$. To do this, for any fixed sequence of $t$ in-context examples $\{\boldsymbol{x}_i, y_i\}_{i=1}^t$, we sample $T \gg d$ query examples $\tilde{\boldsymbol{x}}_k \overset{\text{i.i.d.}}{\sim} \mathcal{N}(\boldsymbol{0}, \boldsymbol{\Sigma})$, where $k \in [T]$. For this fixed sequence of in-context examples $\{\boldsymbol{x}_i, y_i\}_{i=1}^t$, we create $T$ in-context prediction tasks and use the algorithm $\mathcal{A}$ to make predictions $\mathcal{A}(\tilde{\boldsymbol{x}}_k \mid \{\boldsymbol{x}_i, y_i\}_{i=1}^t)$. We define the induced data matrix and labels as

$$\tilde{\boldsymbol{X}} = \begin{bmatrix} \tilde{\boldsymbol{x}}_1^\top \\ \vdots \\ \tilde{\boldsymbol{x}}_T^\top \end{bmatrix} \qquad \tilde{\boldsymbol{Y}} = \begin{bmatrix} \mathcal{A}(\tilde{\boldsymbol{x}}_1 \mid \{\boldsymbol{x}_i, y_i\}_{i=1}^t) \\ \vdots \\ \mathcal{A}(\tilde{\boldsymbol{x}}_T \mid \{\boldsymbol{x}_i, y_i\}_{i=1}^t) \end{bmatrix}. \tag{7}$$

The induced weight vector for $\mathcal{A}$ and these $t$ examples is:

$$\tilde{\boldsymbol{w}}_t(\mathcal{A}) := \tilde{\boldsymbol{w}}_t(\mathcal{A} \mid \{\boldsymbol{x}_i, y_i\}_{i=1}^t) = (\tilde{\boldsymbol{X}}^\top \tilde{\boldsymbol{X}})^{-1} \tilde{\boldsymbol{X}}^\top \tilde{\boldsymbol{Y}}. \tag{8}$$

The similarity of induced weights between two algorithms $\mathcal{A}_a$ and $\mathcal{A}_b$ is the expected average cosine similarity[3] of induced weights $\tilde{\boldsymbol{w}}_t(\mathcal{A}_a)$ and $\tilde{\boldsymbol{w}}_t(\mathcal{A}_b)$ over all possible $1 \le t \le n$, on a randomly sampled data sequence:

$$\text{SimW}(\mathcal{A}_a, \mathcal{A}_b) = \underset{\{\boldsymbol{x}_i, y_i\}_{i=1}^n \sim P_{\mathcal{D}}}{\mathbb{E}} \left[ \frac{1}{n} \sum_{t=1}^n \mathcal{C}\Big( \tilde{\boldsymbol{w}}_t(\mathcal{A}_a | \{\boldsymbol{x}_i, y_i\}_{i=1}^t), \tilde{\boldsymbol{w}}_t(\mathcal{A}_b | \{\boldsymbol{x}_i, y_i\}_{i=1}^t)) \Big) \right].$$

**Matching steps between algorithms.** Each algorithm converges to its predictions after several **steps** — for example the number of iterations for Iterative Newton and GD, and the number of layers for Transformers (see Section 4.1). When comparing two algorithms, given a choice of steps for the first algorithm, we match it with the steps for the second algorithm that maximize similarity.

**Definition 3.4** (Best-matching Steps). Let $\mathcal{M}$ be the metric for evaluating similarities between two algorithms $\mathcal{A}_a$ and $\mathcal{A}_b$, which have steps $p_a \in [0, T_a]$ and $p_b \in [0, T_b]$, respectively. For a given choice of $p_a$, we define the best-matching number of steps of algorithm $\mathcal{A}_b$ for $\mathcal{A}_a$ as:

$$p_b^{\mathcal{M}}(p_a) := \underset{p_b \in [0, T_b]}{\arg\max} \, \mathcal{M}(\mathcal{A}_a(\cdot \mid p_a), \mathcal{A}_b(\cdot \mid p_b)). \tag{9}$$

In our experiments, we chose $T_a, T_b$ be large enough integers so the algorithms converge. The matching processes can be visualized as heatmaps as shown in Figure 3, where best-matching steps are highlighted. This enables us to compare the rate of convergence of algorithms. In particular, if two algorithms converge at the same rate, the best matching steps between the two algorithms should follow a linear trend. We will discuss these results in §4. See Figure 26 on how best-matching steps help compare the convergence rates.

## 4 Experimental Evidence

We primarily study the Transformers-based GPT-2 model with 12 layers and 8 heads per layer. Alternative configurations with fewer heads per layer, or with more layers, also support our findings; we defer them to §A.4.1 and §A.4.2. We initially focus on isotropic cases where $\boldsymbol{\Sigma} = \boldsymbol{I}$ and later consider ill-conditioned $\boldsymbol{\Sigma}$ in §4.3. Our training setup is exactly the same as Garg et al. [2022]: models are trained with at most $n = 40$ in-context examples for $d = 20$ (with the same learning rate, batch size etc.).

We claim that Transformers learn high-order optimization methods in-context. We provide evidence that Transformers improve themselves with more layers in §4.1; Transformers share the same rate of convergence as Iterative Newton, exponentially faster than that of GD, in §4.2; and they also perform well on ill-conditioned problems in §4.3. Finally, we contrast Transformers with LSTMs in §4.5.

---

[3]Alternative metrics such as $\ell_2$ distance gives the same observation. Here cosine similarity is better since errors usually have small magnitudes, and directions of induced weights are meaningful.

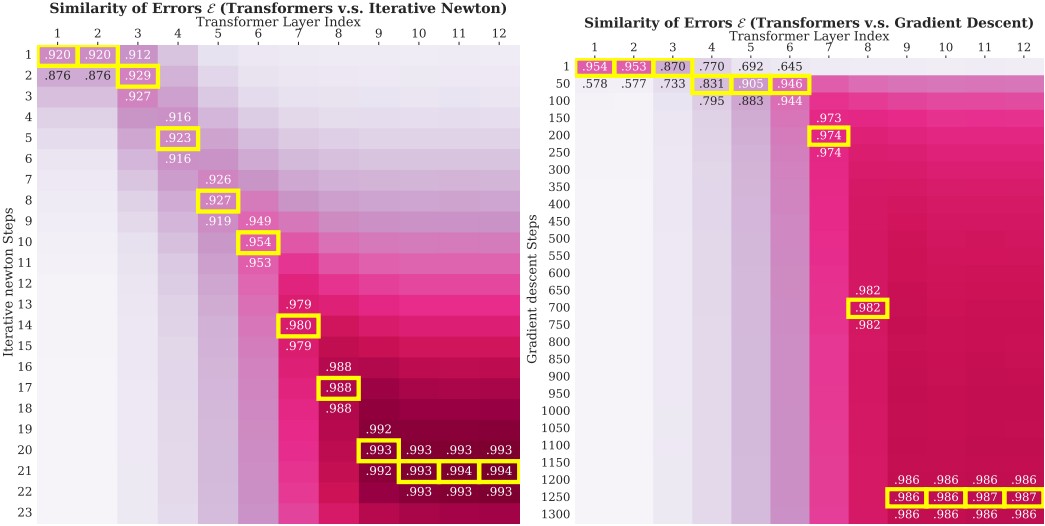

Figure 3: **Heatmaps of Similarity.** The best matching steps are highlighted in yellow. Transformers layers show a linear trend with Iterative Newton steps but an exponential trend with GD. This suggests Transformers and Iterative Newton have the same convergence rate that is exponentially faster than GD. See Figure 10 for an additional heatmap where GD's steps are shown in log scale: on that plot there is a linear correspondence between Transformers and GD's steps. This further strengthens the claim that Transformers have an exponentially faster rate of convergence than GD.

## 4.1 Transformers improve progressively over layers

Many known algorithms for linear regression, including GD, OGD, and Iterative Newton, are *iterative*: their performance progressively improves as they perform more iterations, eventually converging to a final solution. How can a Transformer implement such an iterative algorithm? von Oswald et al. [2022] propose that deeper *layers* of the Transformer may correspond to more iterations; in particular, they show that there exist Transformer parameters such that each attention layer performs one step of GD.

Following this intuition, we first investigate whether the predictions of a trained Transformer improve as the layer index $\ell$ increases. For each layer of hidden states $\boldsymbol{H}^{(\ell)}$ (see Definition 3.2), we re-train the ReadOut to predict $y_t$ for each $t$; the new predictions are given by $\mathrm{ReadOut}^{(\ell)}\left[\boldsymbol{H}^{(\ell)}\right]$. Thus for each input prompt, there are $L$ Transformer predictions parameterized by layer index $\ell$. All parameters besides the ReadOut layer parameters are kept frozen.

As shown in Figure 2(a) (and Figure 7(a) in the Appendix), as we increase the layer index $\ell$, the prediction performance improves progressively. Hence, Transformers progressively improve their predictions over layers $\ell$, similar to how iterative algorithms improve over steps. Such observations are consistent with language tasks where Transformers-based language models also improve their predictions along with layer progressions [Geva et al., 2022, Chuang et al., 2023].

## 4.2 Transformers are more similar to second-order methods, such as Iterative Newton

We now test the more specific hypothesis that the iterative updates performed across Transformer layers are similar to the iterative updates for known iterative algorithms. First, Figure 2 shows that the middle layers of Transformers converge at a rate similar to Iterative Newton, and faster than GD. In particular, the Transformer and Iterative Newton both converge at a superlinear rate, while GD converges at a sublinear rate.

Next, we analyze whether each layer $\ell$ of the Transformer corresponds to performing $k$ steps of some iterative algorithm, for some $k$ depending on $\ell$. We focus here on GD and Iterative Newton's Method; we will discuss online algorithms in Section 4.5, and additional optimization methods in Appendix A.2.3. We will discuss results on noisy linear regression tasks in Appendix A.3.2.

For each layer $\ell$ of the Transformer, we measure the best-matching similarity (see Def. 3.4) with candidate iterative algorithms with the optimal choice of the number of steps $k$. As shown in Figure 3,

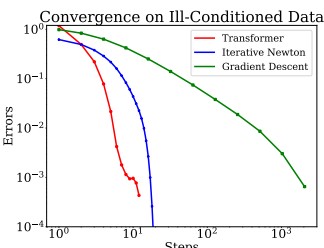

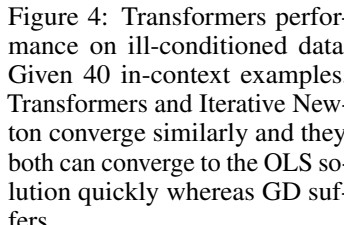

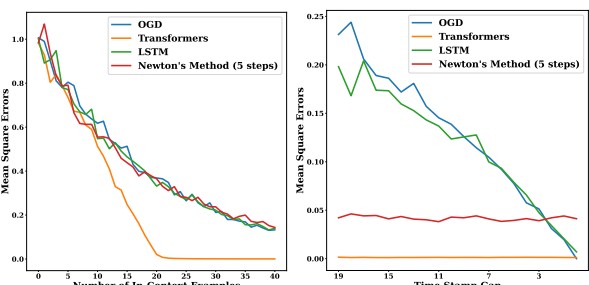

Figure 4: Transformers performance on ill-conditioned data. Given 40 in-context examples, Transformers and Iterative Newton converge similarly and they both can converge to the OLS solution quickly whereas GD suffers.

Figure 5: In the left figure, we measure model predictions with normalized MSE. Though LSTM is seemingly most similar to Newton's Method with only 5 steps, neither algorithm converges yet. OGD also has a similar trend as LSTM. In the right figure, we measure the model's error rate on example $x_{n-g}$ after seeing $n$ examples, for different values of the time stamp gap $g$ (see Appendix A.6), and find both Transformers and not-converged Newton have better memorization than LSTM and OGD.

the Transformer has very high error similarity with Iterative Newton's method at all layers. Moreover, we see a clear *linear* trend between layer 3 and layer 9 of the Transformer, where each layer appears to compute roughly 3 additional iterations of Iterative Newton's method. This trend only stops at the last few layers because both algorithms converge to the OLS solution; Newton is known to converge to OLS (see §3.1), and we verify in Appendix A.2 that the last few layers of the Transformer also basically compute OLS (see Figure 14 in the Appendix). We observe the same trends when using similarity of induced weights as our similarity metric (see Figure 9 in the Appendix). Figure 11 in the Appendix shows that there is a similar *linear* trend between Transformer and BFGS, an alternative quasi-Newton method. This is perhaps not surprising, given that BFGS also gets a superlinear convergence rate for linear regression Nocedal and Wright [1999]. Thus, we do not claim that Transformers specifically implement Iterative Newton, only that they (approximately) implement some second-order method.

In contrast, even though GD has a comparable similarity with the Transformers at later layers, their best matching follows an *exponential* trend. As discussed in the Section 3.1, for well-conditioned problems where $\kappa \approx 1$, to achieve $\epsilon$ error, the rate of convergence of GD is $\mathcal{O}(\log(1/\epsilon))$ while the rate of convergence of Iterative Newton is $\mathcal{O}(\log\log(1/\epsilon))$. Therefore the rate of convergence of Iterative Newton is exponentially faster than GD. Transformer's *linear* correspondence with Iterative Newton and its *exponential* correspondence with GD provides strong evidence that the rate of convergence of Transformers is similar to Iterative Newton, i.e., $\mathcal{O}(\log\log(1/\epsilon))$. We also note that it is not possible to significantly improve GD's convergence rate without using second-order methods: Nemirovski and Yudin [1983] showed a $\Omega(\log(1/\epsilon))$ lower bound on the convergence rate of gradient-based methods for smooth and strongly convex problems, and Arjevani et al. [2016] shows a similar lower bound specifically for quadratic problems. In the Appendix, we show that limited-memory BFGS Liu and Nocedal [1989] and conjugate gradient (see Figure 12), which do not use full-second order information, also converge slower than Transformers. This provides further evidence for the usage of second-order information by Transformers. We also show more evidence by investigating alternative function classes such as linear regression with noises in Appendix A.3.2 and 2-layer neural network with ReLU or Tanh activation function in Appendix A.3.3.

Overall, we conclude that a Transformer trained to perform in-context linear regression learns to implement an algorithm that is very similar to second-order methods, such as Iterative Newton's method, not GD. Starting at layer 3, subsequent layers of the Transformer compute more and more iterations of Iterative Newton's method. This algorithm successfully solves the linear regression problem, as it converges to the optimal OLS solution in the final layers.

### 4.3 Transformers perform well on ill-conditioned data

We repeat the same experiments with data $x_i \overset{\text{i.i.d.}}{\sim} \mathcal{N}(0, \Sigma)$ sampled from an ill-condition covariance matrix $\Sigma$ with condition number $\kappa(\Sigma) = 100$, and eigenbasis chosen uniformly at random. The first

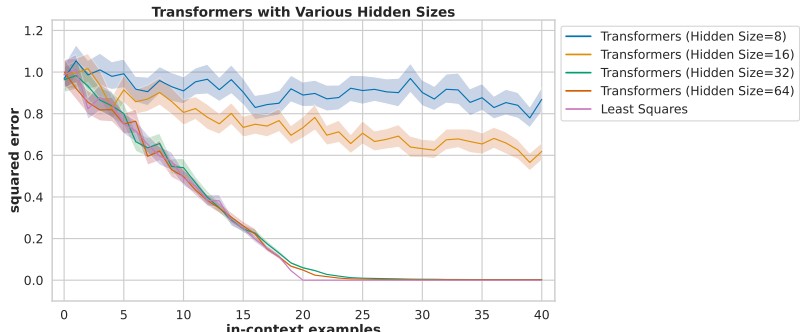

Figure 6: Ablation on Transformer's Hidden Size. For linear regression problems with $d = 20$, Transformers need $\mathcal{O}(d)$ hidden dimension to mimic OLS solutions.

$d/2$ eigenvalues of $\Sigma$ are 100, and the last $d/2$ are 1. Note that choosing the eigenbasis uniformly at random for *each* sequence ensures that there is a different covariance matrix $\Sigma$ for each sequence of datapoints.

As shown in Figure 4, the Transformer model's performance still closely matches Iterative Newton's Method with 21 iterations, same as when $\Sigma = I$ (see layer 10-12 in Figure 3). The convergence of second-order methods has a mild logarithmic dependence on the condition number since they correct for the curvature. On the other hand, GD's convergence is affected polynomially by conditioning. As $\kappa(\Sigma)$ increase from 1 to 100, the number steps required for GD's convergence increases significantly (see Fig. 4 where GD requires 2,000 steps to converge), making it impossible for a 12-layer Transformers to implement these many gradient updates. We also note that preconditioning the data by $(X^\top X)^\dagger$ can make the data well-conditioned, but since the eigenbasis is chosen uniformly at random, with high probability there is no sparse pre-conditioner or any fixed pre-conditioner which works across the data distribution. Computing $(X^\top X)^\dagger$ appears to be as hard as computing the OLS solution (Eq. 1)—in fact Sharan et al. [2019] conjecture that first-order methods such as gradient descent and its variants cannot avoid polynomial dependencies in condition number in the ill-conditioned case.[4] See Appendix A.3.1 for detailed experiments on ill-conditioned problems. These experiments further strengthen our thesis that Transformers learn to perform second-order optimization methods in-context, not first-order methods such as GD.

### 4.4 Transformers Require $\mathcal{O}(d)$ Hidden Dimension

We ablate 12-layer 1-head Transformers with various hidden sizes on $d = 20$ problems. As shown in Figure 6, we observe that Transformers can mimic OLS solution when the hidden size is 32 or 64, but fail with smaller sizes. This resonates with our theoretical results on $\mathcal{O}(d)$ hidden dimension in Theorem 5.1, and in this case, the theorem ensures a construction of transformers to implement Iterative Newton's method.

### 4.5 LSTM is more similar to OGD than Transformers

As discussed in §A.1, LSTM is an alternative auto-regressive model widely used before the introduction of Transformers. Thus, a natural research question is: *If Transformers can learn in-context, can LSTMs do so as well? If so, do they learn the same algorithms?* To answer this question, we train a LSTM model in an identical manner to the Transformers studied in the previous sections.

Figure 5 plots the error of Transformers, LSTMs, and other standard methods as a function of the number of in-context (i.e., training) examples provided. While LSTMs can also learn linear regression in-context, they have much higher mean-squared error than Transformers. Their error rate is similar to Iterative Newton's Method after only 5 iterations, a point where it is far from converging to the OLS solution. Finally, we show that LSTMs behave more like an online learning algorithm than Transformers. In particular, its predictions are biased towards getting more recent training examples correct, as opposed to earlier examples, as shown in Figure 5. This property makes LSTMs similar to

---

[4]Regarding preconditioning, we also note that—even for well-conditioned instances—preconditioned GD still gets a linear rate of convergence, whereas Transformers and Iterative Newton get superlinear rates.

online GD. In contrast, five steps of Newton's method has the same error on average for recent and early examples, showing that the LSTM implements a very different algorithm from a few iterations of Newton. We hypothesize that since LSTMs have limited memory, they must learn in a roughly online fashion; in contrast, Transformer's attention heads can access the entire sequence of past examples, enabling it to learn more complex algorithms. See §A.1 for more discussions.

## 5 Theoretical Justification

Our empirical evidence demonstrates that Transformers behave much more similarly to Iterative Newton's than to GD. Iterative Newton is a second-order optimization method, and is algorithmically more involved than GD. We begin by first examining this difference in complexity. As discussed in Section 3, the updates for Iterative Newton are of the form,

$$\hat{\boldsymbol{w}}_{k+1}^{\text{Newton}} = \boldsymbol{M}_{k+1} \boldsymbol{X}^\top \boldsymbol{y} \qquad \text{where } \boldsymbol{M}_{k+1} = 2\boldsymbol{M}_k - \boldsymbol{M}_k \boldsymbol{S} \boldsymbol{M}_k \tag{10}$$

and $\boldsymbol{M}_0 = \alpha \boldsymbol{S}$ for some $\alpha > 0$. We can express $\boldsymbol{M}_k$ in terms of powers of $\boldsymbol{S}$ by expanding iteratively, for example $\boldsymbol{M}_1 = 2\alpha \boldsymbol{S} - 4\alpha^2 \boldsymbol{S}^3$, $\boldsymbol{M}_2 = 4\alpha \boldsymbol{S} - 12\alpha^2 \boldsymbol{S}^3 + 16\alpha^3 \boldsymbol{S}^5 - 16\alpha^4 \boldsymbol{S}^7$, and in general $\boldsymbol{M}_k = \sum_{s=1}^{2^{k+1}-1} \beta_s \boldsymbol{S}^s$ for some $\beta_s \in \mathbb{R}$ (see Appendix B.3 for detailed calculations). Note that $k$ steps of Iterative Newton's requires computing $\Omega(2^k)$ moments of $\boldsymbol{S}$. Let us contrast this with GD. GD updates for linear regression take the form,

$$\hat{\boldsymbol{w}}_{k+1}^{\text{GD}} = \hat{\boldsymbol{w}}_k^{\text{GD}} - \eta(\boldsymbol{S}\hat{\boldsymbol{w}}_k^{\text{GD}} - \boldsymbol{X}^\top \boldsymbol{y}). \tag{11}$$

Like Iterative Newton, we can express $\hat{\boldsymbol{w}}_k^{\text{GD}}$ in terms of powers of $\boldsymbol{S}$ and $\boldsymbol{X}^\top \boldsymbol{y}$. However, after $k$ steps of GD, the highest power of $\boldsymbol{S}$ is only $O(k)$. This exponential separation is consistent with the exponential gap in terms of the parameter dependence in the convergence rate—$\mathcal{O}\left(\kappa(\boldsymbol{S}) \log(1/\epsilon)\right)$ for GD vs. $\mathcal{O}(\log \kappa(\boldsymbol{S}) + \log\log(1/\epsilon))$ for Iterative Newton. Therefore, a natural question is whether Transformers can actually as complicated of a method such as Iterative Newton with only polynomially many layers? Theorem 5.1 shows that this is indeed possible.

**Theorem 5.1.** *For any $k$, there exist Transformer weights such that on any set of in-context examples $\{\boldsymbol{x}_i, y_i\}_{i=1}^n$ and test point $\boldsymbol{x}_{\text{test}}$, the Transformer predicts on $\boldsymbol{x}_{\text{test}}$ using $\boldsymbol{x}_{\text{test}}^\top \hat{\boldsymbol{w}}_k^{\text{Newton}}$. Here $\hat{\boldsymbol{w}}_k^{\text{Newton}}$ are the Iterative Newton updates given by $\hat{\boldsymbol{w}}_k^{\text{Newton}} = \boldsymbol{M}_k \boldsymbol{X}^\top \boldsymbol{y}$ where $\boldsymbol{M}_j$ is updated as*

$$\boldsymbol{M}_j = 2\boldsymbol{M}_{j-1} - \boldsymbol{M}_{j-1} \boldsymbol{S} \boldsymbol{M}_{j-1}, 1 \le j \le k, \quad \boldsymbol{M}_0 = \alpha \boldsymbol{S},$$

*for some $\alpha > 0$ and $\boldsymbol{S} = \boldsymbol{X}^\top \boldsymbol{X}$. The dimensionality of the hidden layers is $\mathcal{O}(d)$, and the number of layers is $k + 8$. One transformer layer computes one Newton iteration. 3 initial transformer layers are needed for initializing $\boldsymbol{M}_0$ and 5 layers at the end are needed to read out predictions from the computed pseudo-inverse $\boldsymbol{M}_k$.*

We note that our proof uses full attention instead of causal attention and ReLU activations for the self-attention layers. The definitions of these and the full proof appear in Appendix B.

## 6 Conclusion and Discussion

In this work, we studied how Transformers perform in-context learning for linear regression. In contrast with the hypothesis that Transformers learn in-context by implementing gradient descent, our experimental results show that different Transformer layers match iterations of Iterative Newton *linearly* and Gradient Descent *exponentially*. This suggests that Transformers share a similar rate of convergence to Iterative Newton but not to Gradient Descent. Moreover, Transformers can perform well empirically on ill-conditioned linear regression, whereas first-order methods such as Gradient Descent struggle. This empirical evidence — when combined with existing lower bounds in optimization — suggests that Transformers use second-order information for solving linear regression, and we also prove that Transformers can indeed represent second-order methods.

An interesting direction is to explore a wider range of second-order methods that Transformers can implement. It also seems promising to extend our analysis to classification problems, especially given recent work showing that Transformers resemble SVMs in classification tasks [Li et al., 2023, Tarzanagh et al., 2023a]. Finally, a natural question is to understand the differences in the model architecture that make Transformers better in-context learners than LSTMs. Based on our investigations with LSTMs, we hypothesize that Transformers can implement more powerful algorithms because of having access to a longer history of examples. Investigating the role of this additional memory in learning appears to be an intriguing direction.

# Acknowledgement

We would like to thank the USC NLP Group and Center for AI Safety for providing compute resources. DF would like to thank Oliver Liu and Ameya Godbole for their extensive discussions. DF and RJ were supported by a Google Research Scholar Award. RJ was also supported by an Open Philanthropy research grant. VS was supported by NSF CAREER Award CCF-2239265 and an Amazon Research Award.

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

# Appendix

## A  Additional Experimental Results

### A.1  Contrast with LSTMs

While our primary goal is to analyze Transformers, we also consider LSTMs [Hochreiter and Schmidhuber, 1997] to understand whether Transformers learn different algorithms than other neural sequence models trained to do linear regression. In particular, we train a unidirectional $L$-layer LSTM, which generates a sequence of hidden states $\boldsymbol{H}^{(\ell)}$ for each layer $\ell$, similarly to an $L$-layer Transformer. As with Transformers, we add a readout layer that predicts the $\hat{y}_{t+1}^{\text{LSTM}}$ from the final hidden state at the final layer, $\boldsymbol{H}_{:,2t+1}^{(L)}$.

|        | Transformers | LSTM  |
|--------|:------------:|:-----:|
| Newton |  **0.991**   | 0.920 |
| GD     |  **0.957**   | 0.916 |
| OGD    |    0.806     | **0.954** |

Table 1: **Similarity of errors between algorithms.** Transformers are more similar to full-observation methods such as Newton and GD; and LSTMs are more similar to online methods such as OGD.

We train a 10-layer LSTM model, with 5.3M parameters, in an identical manner to the Transformers (with 9.5M parameters) studied in the previous sections.[5]

LSTMs' inferior performance to Transformers can be explained by the inability of LSTMs to use deeper layers to improve their predictions. Figure 7 shows that LSTM performance does not improve across layers—a readout head fine-tuned for the first layer makes equally good predictions as the full 10-layer model. Thus, LSTMs seem poorly equipped to fully implement iterative algorithms. Similarly, Table 1 shows that LSTMs are more similar to OGD than Transformers are, whereas Transformers are more similar to Newton and GD than LSTMs.

## A.2 Additional Results on Isotropic Data without Noise

### A.2.1 Progression of Algorithms

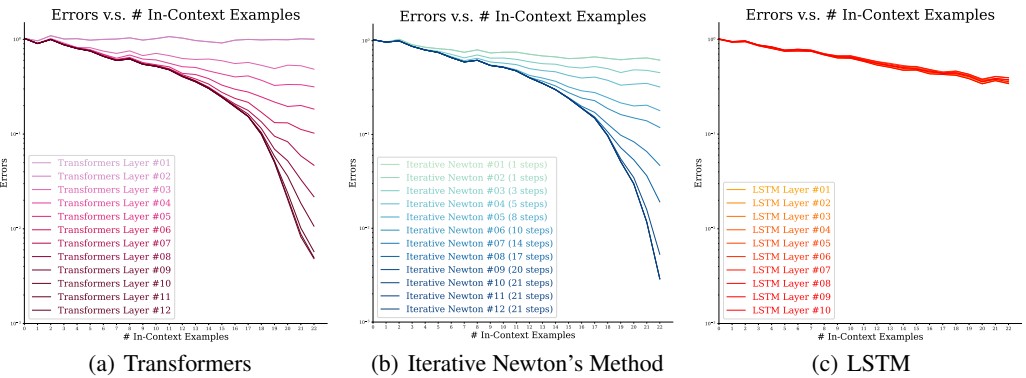

(a) Transformers      (b) Iterative Newton's Method      (c) LSTM

Figure 7: **Progression of Algorithms.** (a) Transformer's performance improves over the layer index $\ell$. (b) Iterative Newton's performance improves over the number of iterations $k$, in a way that closely resembles the Transformer. We plot the best-matching $k$ to Transformer's $\ell$ following Definition 3.4. (c) In contrast, LSTM's performance does not improve from layer to layer.

### A.2.2 Heatmaps

We present heatmaps with all values of similarities.

---

[5]While the LSTM has fewer parameters than the Transformer, we found in preliminary experiments that increasing the size of the LSTM would not substantively change our results.

**Similarity of Errors $\mathcal{E}$ (Transformers v.s. Iterative Newton)**

Trasformer Layer Index

| Iterative newton Steps | 1 | 2 | 3 | 4 | 5 | 6 | 7 | 8 | 9 | 10 | 11 | 12 |
|---|---|---|---|---|---|---|---|---|---|---|---|---|
| 1 | .920 | .920 | .912 | .816 | .716 | .662 | .634 | .623 | .618 | .613 | .620 | .616 |
| 2 | .876 | .876 | .929 | .858 | .760 | .702 | .672 | .660 | .655 | .651 | .656 | .652 |
| 3 | .829 | .828 | .927 | .893 | .805 | .745 | .713 | .700 | .694 | .690 | .695 | .692 |
| 4 | .781 | .781 | .911 | .916 | .848 | .789 | .755 | .741 | .735 | .732 | .735 | .733 |
| 5 | .734 | .734 | .883 | .923 | .886 | .834 | .798 | .783 | .777 | .774 | .777 | .774 |
| 6 | .691 | .691 | .850 | .916 | .912 | .875 | .840 | .824 | .817 | .814 | .817 | .815 |
| 7 | .652 | .652 | .814 | .898 | .926 | .910 | .878 | .862 | .855 | .852 | .854 | .852 |
| 8 | .619 | .619 | .779 | .874 | .927 | .935 | .911 | .895 | .888 | .885 | .886 | .885 |
| 9 | .591 | .591 | .748 | .849 | .919 | .949 | .937 | .921 | .915 | .913 | .913 | .913 |
| 10 | .569 | .569 | .723 | .826 | .907 | .954 | .956 | .942 | .936 | .934 | .935 | .934 |
| 11 | .552 | .552 | .703 | .807 | .894 | .953 | .968 | .958 | .953 | .950 | .951 | .950 |
| 12 | .539 | .539 | .688 | .792 | .882 | .949 | .976 | .969 | .965 | .962 | .962 | .962 |
| 13 | .530 | .530 | .677 | .780 | .871 | .944 | .979 | .977 | .973 | .971 | .971 | .971 |
| 14 | .524 | .524 | .669 | .771 | .863 | .938 | .980 | .983 | .980 | .978 | .978 | .978 |
| 15 | .520 | .519 | .664 | .765 | .857 | .933 | .979 | .986 | .985 | .983 | .983 | .983 |
| 16 | .517 | .517 | .660 | .760 | .852 | .929 | .977 | .988 | .988 | .987 | .986 | .987 |
| 17 | .515 | .515 | .657 | .757 | .848 | .926 | .975 | .988 | .990 | .989 | .989 | .989 |
| 18 | .513 | .513 | .655 | .754 | .846 | .924 | .973 | .988 | .992 | .991 | .991 | .991 |
| 19 | .512 | .512 | .653 | .752 | .843 | .921 | .972 | .988 | .992 | .992 | .992 | .993 |
| 20 | .511 | .511 | .652 | .751 | .842 | .920 | .970 | .987 | .993 | .993 | .993 | .993 |
| 21 | .511 | .511 | .651 | .750 | .840 | .918 | .969 | .986 | .992 | .993 | .994 | .994 |
| 22 | .510 | .510 | .649 | .749 | .839 | .917 | .967 | .984 | .991 | .993 | .993 | .993 |
| 23 | .508 | .508 | .646 | .746 | .835 | .913 | .963 | .981 | .988 | .989 | .990 | .990 |

**Similarity of Errors $\mathcal{E}$ (Transformers v.s. Gradient Descent)**

Trasformer Layer Index

| Gradient descent Steps | 1 | 2 | 3 | 4 | 5 | 6 | 7 | 8 | 9 | 10 | 11 | 12 |
|---|---|---|---|---|---|---|---|---|---|---|---|---|
| 1 | .954 | .953 | .870 | .770 | .692 | .645 | .620 | .610 | .606 | .600 | .607 | .603 |
| 50 | .578 | .577 | .733 | .831 | .905 | .946 | .954 | .946 | .941 | .939 | .939 | .938 |
| 100 | .543 | .543 | .694 | .795 | .883 | .944 | .970 | .967 | .963 | .961 | .962 | .961 |
| 150 | .531 | .531 | .679 | .781 | .871 | .939 | .973 | .974 | .972 | .970 | .970 | .969 |
| 200 | .525 | .524 | .672 | .772 | .863 | .935 | .974 | .977 | .976 | .974 | .974 | .974 |
| 250 | .521 | .521 | .667 | .767 | .858 | .932 | .974 | .979 | .978 | .977 | .977 | .977 |
| 300 | .518 | .518 | .664 | .763 | .855 | .929 | .973 | .980 | .980 | .979 | .979 | .979 |
| 350 | .516 | .516 | .661 | .761 | .852 | .927 | .973 | .981 | .981 | .980 | .980 | .980 |
| 400 | .515 | .515 | .660 | .759 | .850 | .926 | .972 | .982 | .983 | .981 | .981 | .981 |
| 450 | .514 | .514 | .658 | .757 | .849 | .924 | .972 | .982 | .983 | .982 | .982 | .982 |
| 500 | .514 | .514 | .657 | .756 | .847 | .923 | .971 | .982 | .984 | .983 | .983 | .983 |
| 550 | .512 | .512 | .656 | .755 | .846 | .922 | .970 | .982 | .984 | .983 | .983 | .983 |
| 600 | .512 | .512 | .655 | .753 | .845 | .921 | .970 | .982 | .984 | .983 | .984 | .984 |
| 650 | .512 | .511 | .655 | .753 | .844 | .921 | .970 | .982 | .984 | .984 | .984 | .984 |
| 700 | .511 | .511 | .654 | .752 | .844 | .920 | .969 | .982 | .985 | .984 | .985 | .985 |
| 750 | .511 | .511 | .653 | .752 | .843 | .920 | .969 | .982 | .985 | .985 | .985 | .985 |
| 800 | .510 | .510 | .652 | .751 | .842 | .919 | .968 | .982 | .985 | .984 | .985 | .985 |
| 850 | .510 | .510 | .652 | .750 | .841 | .919 | .968 | .982 | .985 | .985 | .985 | .985 |
| 900 | .510 | .510 | .652 | .750 | .841 | .918 | .968 | .982 | .986 | .985 | .986 | .986 |
| 950 | .509 | .509 | .652 | .750 | .841 | .918 | .968 | .982 | .986 | .986 | .986 | .986 |
| 1000 | .509 | .508 | .651 | .749 | .840 | .917 | .967 | .982 | .986 | .985 | .986 | .986 |
| 1050 | .509 | .509 | .651 | .749 | .840 | .917 | .967 | .981 | .986 | .985 | .986 | .986 |
| 1100 | .510 | .509 | .651 | .749 | .840 | .916 | .967 | .981 | .986 | .986 | .986 | .986 |
| 1150 | .509 | .508 | .650 | .748 | .839 | .916 | .966 | .981 | .986 | .986 | .986 | .986 |
| 1200 | .508 | .508 | .650 | .748 | .839 | .916 | .966 | .981 | .986 | .986 | .986 | .986 |
| 1250 | .508 | .508 | .650 | .748 | .839 | .916 | .966 | .981 | .986 | .986 | .987 | .987 |
| 1300 | .508 | .508 | .650 | .748 | .838 | .915 | .966 | .981 | .986 | .986 | .986 | .986 |

Figure 8: **Similarity of Errors.** The best matching steps are highlighted in yellow.

**Similarity of Induced Weight $\tilde{w}$ (Transformers v.s. Iterative Newton)**

Trasformer Layer Index

| Iterative newton Steps | 1 | 2 | 3 | 4 | 5 | 6 | 7 | 8 | 9 | 10 | 11 | 12 |
|---|---|---|---|---|---|---|---|---|---|---|---|---|
| 1 | -.000 | .001 | .859 | .811 | .742 | .719 | .714 | .711 | .711 | .711 | .711 | .712 |
| 2 | .000 | .001 | .872 | .856 | .795 | .769 | .763 | .760 | .760 | .760 | .760 | .761 |
| 3 | .001 | .001 | .870 | .890 | .844 | .816 | .809 | .806 | .806 | .805 | .806 | .806 |
| 4 | .002 | .001 | .857 | .909 | .883 | .857 | .849 | .845 | .845 | .845 | .845 | .846 |
| 5 | .003 | .000 | .838 | .915 | .911 | .889 | .881 | .877 | .877 | .876 | .877 | .877 |
| 6 | .004 | -.000 | .819 | .912 | .928 | .914 | .906 | .902 | .902 | .901 | .902 | .902 |
| 7 | .005 | -.000 | .801 | .903 | .937 | .932 | .926 | .922 | .921 | .921 | .922 | .922 |
| 8 | .005 | -.001 | .785 | .893 | .939 | .944 | .941 | .937 | .936 | .936 | .937 | .937 |
| 9 | .005 | -.001 | .773 | .883 | .936 | .951 | .952 | .948 | .947 | .947 | .948 | .948 |
| 10 | .006 | -.001 | .763 | .874 | .932 | .953 | .959 | .955 | .955 | .955 | .956 | .956 |
| 11 | .006 | -.000 | .756 | .867 | .927 | .954 | .963 | .961 | .961 | .960 | .961 | .961 |
| 12 | .006 | -.000 | .750 | .862 | .923 | .953 | .966 | .965 | .965 | .964 | .965 | .965 |
| 13 | .006 | -.000 | .747 | .858 | .920 | .952 | .967 | .967 | .968 | .967 | .968 | .968 |
| 14 | .006 | .000 | .744 | .855 | .918 | .950 | .968 | .969 | .969 | .969 | .970 | .970 |
| 15 | .006 | .000 | .742 | .853 | .916 | .949 | .967 | .970 | .971 | .970 | .971 | .971 |
| 16 | .006 | .000 | .741 | .851 | .914 | .948 | .967 | .970 | .972 | .971 | .972 | .972 |
| 17 | .007 | .000 | .740 | .850 | .913 | .947 | .966 | .970 | .972 | .972 | .973 | .973 |
| 18 | .007 | .000 | .739 | .849 | .912 | .946 | .966 | .970 | .973 | .972 | .973 | .974 |
| 19 | .007 | .000 | .739 | .849 | .911 | .945 | .966 | .970 | .973 | .973 | .974 | .974 |
| 20 | .007 | .000 | .738 | .848 | .911 | .945 | .965 | .970 | .973 | .973 | .974 | .974 |
| 21 | .007 | .000 | .738 | .848 | .911 | .944 | .965 | .970 | .973 | .973 | .974 | .974 |
| 22 | .007 | .000 | .738 | .848 | .910 | .944 | .965 | .970 | .973 | .973 | .974 | .974 |
| 23 | .007 | .000 | .738 | .848 | .910 | .944 | .965 | .970 | .973 | .973 | .974 | .974 |

**Similarity of Induced Weight $\tilde{w}$ (Transformers v.s. Gradient Descent)**

Trasformer Layer Index

| Gradient descent Steps | 1 | 2 | 3 | 4 | 5 | 6 | 7 | 8 | 9 | 10 | 11 | 12 |
|---|---|---|---|---|---|---|---|---|---|---|---|---|
| 1 | .069 | -.002 | .771 | .731 | .695 | .683 | .674 | .671 | .671 | .669 | .670 | .670 |
| 50 | .020 | .004 | .772 | .880 | .934 | .958 | .965 | .963 | .964 | .962 | .964 | .964 |
| 100 | .020 | .005 | .757 | .866 | .927 | .959 | .971 | .970 | .971 | .969 | .971 | .971 |
| 150 | .020 | .005 | .752 | .861 | .923 | .958 | .972 | .973 | .974 | .972 | .974 | .974 |
| 200 | .020 | .005 | .749 | .858 | .921 | .957 | .973 | .974 | .975 | .973 | .975 | .975 |
| 250 | .020 | .005 | .748 | .856 | .919 | .956 | .973 | .974 | .976 | .974 | .976 | .976 |
| 300 | .019 | .005 | .747 | .855 | .918 | .955 | .973 | .975 | .976 | .974 | .976 | .976 |
| 350 | .020 | .005 | .746 | .854 | .917 | .954 | .973 | .975 | .976 | .975 | .977 | .977 |
| 400 | .020 | .005 | .745 | .854 | .917 | .954 | .972 | .975 | .977 | .975 | .977 | .977 |
| 450 | .020 | .005 | .745 | .853 | .916 | .953 | .972 | .975 | .977 | .975 | .977 | .977 |
| 500 | .020 | .005 | .744 | .853 | .916 | .953 | .972 | .975 | .977 | .976 | .977 | .977 |
| 550 | .020 | .005 | .744 | .852 | .915 | .953 | .972 | .975 | .977 | .976 | .977 | .977 |
| 600 | .019 | .005 | .744 | .852 | .915 | .953 | .972 | .975 | .977 | .976 | .978 | .977 |
| 650 | .020 | .005 | .744 | .852 | .915 | .952 | .972 | .975 | .977 | .976 | .978 | .978 |
| 700 | .020 | .005 | .743 | .851 | .915 | .952 | .972 | .975 | .977 | .976 | .978 | .978 |
| 750 | .020 | .005 | .743 | .851 | .914 | .952 | .971 | .975 | .977 | .976 | .978 | .978 |
| 800 | .020 | .005 | .743 | .851 | .914 | .952 | .971 | .975 | .977 | .976 | .978 | .978 |
| 850 | .020 | .005 | .743 | .851 | .914 | .952 | .971 | .975 | .977 | .976 | .978 | .978 |
| 900 | .020 | .005 | .743 | .851 | .914 | .952 | .971 | .975 | .977 | .976 | .978 | .978 |
| 950 | .020 | .005 | .743 | .851 | .914 | .952 | .971 | .975 | .977 | .976 | .978 | .978 |
| 1000 | .020 | .005 | .743 | .851 | .914 | .951 | .971 | .975 | .977 | .976 | .978 | .978 |
| 1050 | .020 | .005 | .743 | .851 | .914 | .951 | .971 | .975 | .978 | .976 | .978 | .978 |
| 1100 | .020 | .005 | .742 | .850 | .914 | .951 | .971 | .975 | .978 | .976 | .978 | .978 |
| 1150 | .020 | .005 | .742 | .851 | .914 | .951 | .971 | .975 | .978 | .976 | .978 | .978 |
| 1200 | .020 | .005 | .742 | .850 | .913 | .951 | .971 | .975 | .978 | .976 | .978 | .978 |
| 1250 | .020 | .005 | .742 | .850 | .913 | .951 | .971 | .975 | .978 | .976 | .978 | .978 |
| 1300 | .019 | .005 | .742 | .850 | .913 | .951 | .971 | .975 | .978 | .976 | .978 | .978 |

Figure 9: **Similarity of Induced Weight Vectors.** The best matching steps are highlighted in yellow.

**Similarity of Errors $\mathcal{E}$ (Transformers v.s. Gradient Descent)**

Trasformer Layer Index

| Gradient descent Steps | 1 | 2 | 3 | 4 | 5 | 6 | 7 | 8 | 9 | 10 | 11 | 12 |
|---|---|---|---|---|---|---|---|---|---|---|---|---|
| 1 | .953 | .953 | .870 | .771 | .692 | .645 | .620 | .609 | .605 | .599 | .606 | .602 |
| 2 | .910 | .910 | .903 | .826 | .750 | .703 | .676 | .665 | .660 | .655 | .661 | .657 |
| 4 | .842 | .841 | .913 | .878 | .816 | .773 | .746 | .733 | .728 | .724 | .728 | .725 |
| 8 | .759 | .759 | .886 | .905 | .876 | .846 | .820 | .807 | .801 | .798 | .801 | .799 |
| 16 | .678 | .677 | .831 | .895 | .910 | .903 | .886 | .873 | .867 | .865 | .867 | .865 |
| 32 | .610 | .610 | .768 | .858 | .914 | .938 | .934 | .924 | .918 | .916 | .917 | .916 |
| 64 | .563 | .563 | .717 | .817 | .897 | .947 | .961 | .954 | .950 | .948 | .948 | .947 |
| 128 | .536 | .535 | .685 | .786 | .875 | .941 | .972 | .971 | .968 | .966 | .967 | .966 |
| 256 | .521 | .521 | .666 | .766 | .858 | .932 | .973 | .979 | .978 | .977 | .977 | .977 |
| 512 | .513 | .513 | .656 | .755 | .847 | .923 | .971 | .982 | .984 | .982 | .983 | .983 |
| 1024 | .509 | .509 | .652 | .749 | .840 | .917 | .967 | .982 | .986 | .985 | .986 | .986 |
| 2048 | .507 | .507 | .648 | .745 | .836 | .913 | .964 | .980 | .986 | .986 | .987 | .987 |
| 4096 | .506 | .505 | .646 | .744 | .834 | .911 | .962 | .979 | .985 | .987 | .988 | .988 |

Figure 10: **Similarity of Errors of Gradient Descent in Log Scale.** The best matching steps are highlighted in yellow. Putting the number of steps of Gradient Descent in log scale further verifies the claim that Transformer's rate of covergence is exponentially faster than that of Gradient Descent.

### A.2.3 Comparison with Other Second-Order Methods

In this section, we ablate with alternative second-order methods, such as Conjugate Gradient, BFGS, and its limited memory variant, L-BFGS.

**Conjugate Gradient Method.** For linear regression problems, the Conjugate Gradient (CG) method solves the linear system

$$\underbrace{(\boldsymbol{X}^\top \boldsymbol{X})}_{\boldsymbol{S}} \boldsymbol{w} - \boldsymbol{X}^\top \boldsymbol{y} = 0$$

CG finds the weight vector $\hat{\boldsymbol{w}}^{CG}$ with initialization $\boldsymbol{w}_0$ by maintain a set of conjugate gradient $\{\Delta \boldsymbol{w}_1, \cdots, \Delta \boldsymbol{w}_k\}$. It follows the iterative update rule

$$
\begin{aligned}
\boldsymbol{d}_k &= -\nabla \mathcal{L}(\boldsymbol{w}_k) \\
\Delta \boldsymbol{w}_k &= \boldsymbol{d}_k - \sum_{i=0}^{k-1} \frac{\boldsymbol{d}_k^\top \boldsymbol{S} \Delta \boldsymbol{w}_i}{\Delta \boldsymbol{w}_i^\top \boldsymbol{S} \Delta \boldsymbol{w}_i} \Delta \boldsymbol{w}_i \\
\alpha_k &= \arg\min_\alpha \mathcal{L}(\boldsymbol{w}_k + \alpha \Delta \boldsymbol{w}_k) \\
\boldsymbol{w}_{k+1} &= \boldsymbol{w}_k + \alpha_k \Delta \boldsymbol{w}_k
\end{aligned}
\tag{12}
$$

The conjugate Gradient method requires $\mathcal{O}\left(\sqrt{\kappa} \log(1/\epsilon)\right)$ steps to converge to an $\epsilon$ error on quadratic objectives such as linear regression.

**BFGS.** Broyden– Fletcher–Goldfarb–Shanno (BFGS) is a Quasi-Newton method, designed to approximate the inverse Hessian $\boldsymbol{B}_k :\approx \nabla^2 \mathcal{L}(\boldsymbol{w}_k)^{-1}$. The BFGS updates are given by

$$\boldsymbol{w}_{k+1} = \boldsymbol{w}_k - \alpha_k \boldsymbol{B}_k \nabla \mathcal{L}(\boldsymbol{w}_k) \tag{13}$$

where

$$
\begin{aligned}
\boldsymbol{s}_k &= \boldsymbol{w}_{k+1} - \boldsymbol{w}_k \\
\boldsymbol{y}_k &= \nabla \mathcal{L}(\boldsymbol{w}_{k+1}) - \nabla \mathcal{L}(\boldsymbol{w}_k) \\
\boldsymbol{B}_{k+1} &= \boldsymbol{B}_k - \frac{\boldsymbol{B}_k \boldsymbol{y}_k \boldsymbol{y}_k^\top \boldsymbol{B}_k}{\boldsymbol{y}_k^\top \boldsymbol{B}_k \boldsymbol{y}_k} + \frac{\boldsymbol{s}_k \boldsymbol{s}_k^\top}{\boldsymbol{y}_k^\top \boldsymbol{s}_k}
\end{aligned}
$$

When $k$ is large, $\boldsymbol{B}_k$ approximates the inverse Hessian well.

**L(imited-memory)-BFGS.** L-BFGS is a limited-memory version of BFGS. Instead of the inverse Hessian $\boldsymbol{B}_k$, L-BFGS maintains a history of past $m$ updates (where $m$ is usually small). Recall the iterative update rule of $\boldsymbol{B}_k$ in BFGS

$$\boldsymbol{B}_{k+1} = \boldsymbol{B}_k - \frac{\boldsymbol{B}_k \boldsymbol{y}_k \boldsymbol{y}_k^\top \boldsymbol{B}_k}{\boldsymbol{y}_k^\top \boldsymbol{B}_k \boldsymbol{y}_k} + \frac{\boldsymbol{s}_k \boldsymbol{s}_k^\top}{\boldsymbol{y}_k^\top \boldsymbol{s}_k} \tag{14}$$

Unlike BFGS, which recursively unroll to an initialization $\boldsymbol{B}_0$, L-BFGS only unroll to $\boldsymbol{B}_{k-m}$ but replacing $\boldsymbol{B}_{k-m}$ with $\boldsymbol{B}_{\text{init}}$. In this regard, running $n$ steps of L-BFGS only requires $\mathcal{O}(mn)$ memory, which is more memory-efficient than BFGS who requires $\mathcal{O}(n^2)$ memory. The trade-off is that L-BFGS won't have a good estimate of the inverse Hessian when $m < d$, where $d$ is the dimensionality of the quadratic problem. In this regard, it will converge slower than full BFGS.

In Figure 11 and Figure 12, we compare Transformers with BFGS, L-BFGS, and Conjugate Gradient method on the metric of similarity of errors. We find that Transformers have a similar *linear* correspondence with BFGS. This is perhaps not surprising, given that BFGS also gets a superlinear convergence rate for linear regression Nocedal and Wright [1999]. Meanwhile, Transformers show a substantially faster convergence rate than L-BFGS and CG.

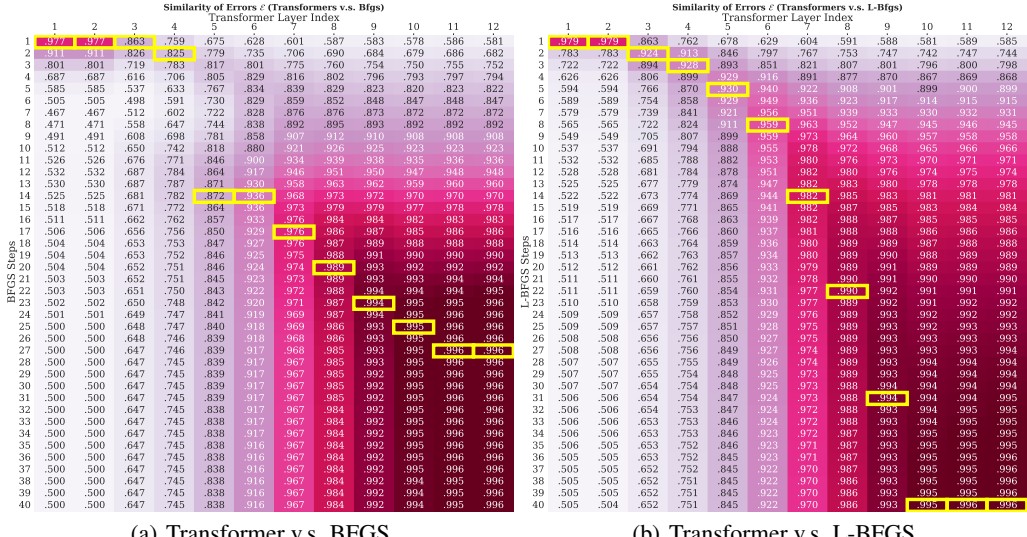

(a) Transformer v.s. BFGS

(b) Transformer v.s. L-BFGS

Figure 11: **Similarity of Errors between Transformers and BFGS or L-BFGS.** The best matching steps are highlighted in yellow. We find that Transformer, from layers 6 to 11, has a linear correspondence with BFGS. For L-BFGS, due to its limited memory, it approximates second-order information more slowly and results in a slower convergence rate than Transformers.

Figure 12: **Similarity of Errors between Transformers and Conjugate Gradient.** Transformer's convergence rate is still faster than conjugate gradient methods.

### A.2.4 Additional Results on Comparison over Transformer Layers

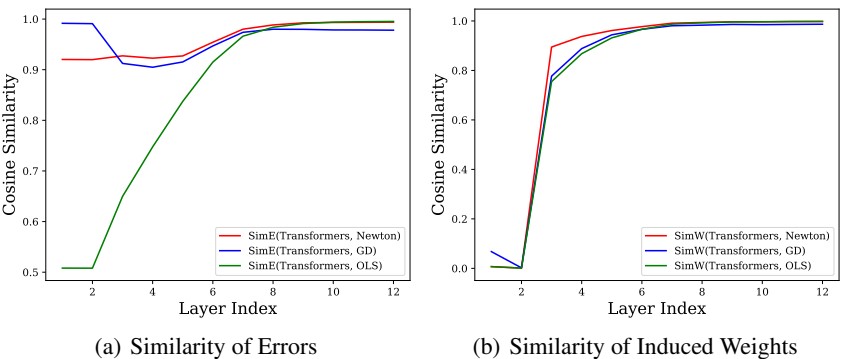

(a) Similarity of Errors      (b) Similarity of Induced Weights

Figure 13: Similarities between Transformer and candidate algorithms. Transformers resemble *Iterative Newton's Method* the most.

### A.2.5 Additional Results on Similarity of Induced Weights

We present more details line plots for how the similarity of weights changes as the models see more in-context observations $\{\boldsymbol{x}_i, y_i\}_{i=1}^n$, i.e., as $n$ increases. We fix the number of Transformers layers $\ell$ and compare with other algorithms with their best-match steps to $\ell$ in Figure 14.

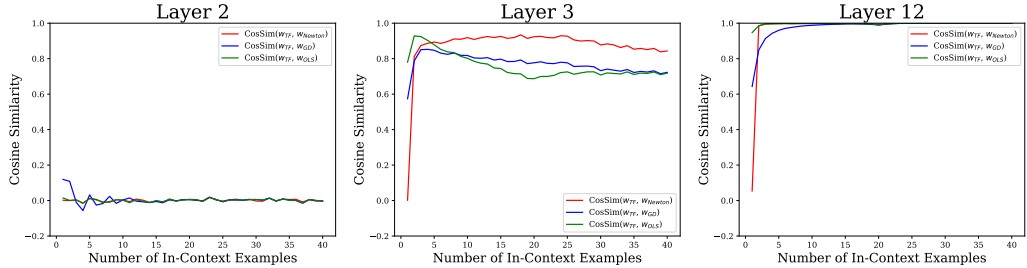

Figure 14: *Similarity of induced weights* over varying number of in-context examples, on three layer indices of Transformers, indexed as 2, 3 and 12. We find that initially at layer 2, the Transformers model hasn't learned so it has zero similarity to all candidate algorithms. As we progress to the next layer number 3, we find that Transformers start to learn, and when provided few examples, Transformers are more similar to OLS but soon become most similar to the Iterative Newton's Method. Layer 12 shows that Transformers in the later layers converge to the OLS solution when provided more than 1 example. We also find there is a dip around $n = d$ for similarity between Transformers and OLS but not for Transformers and Newton, and this is probably because OLS has a more prominent double-descent phenomenon than Transformers and Newton.

### A.3.1  Experiments on Ill-Conditioned Problems

In this section, we repeat the same experiments as we did on isotropic data in the main text and in Appendix A.2, and we change the covariance matrix to be ill-conditioned such that $\kappa(\Sigma) = 100$.

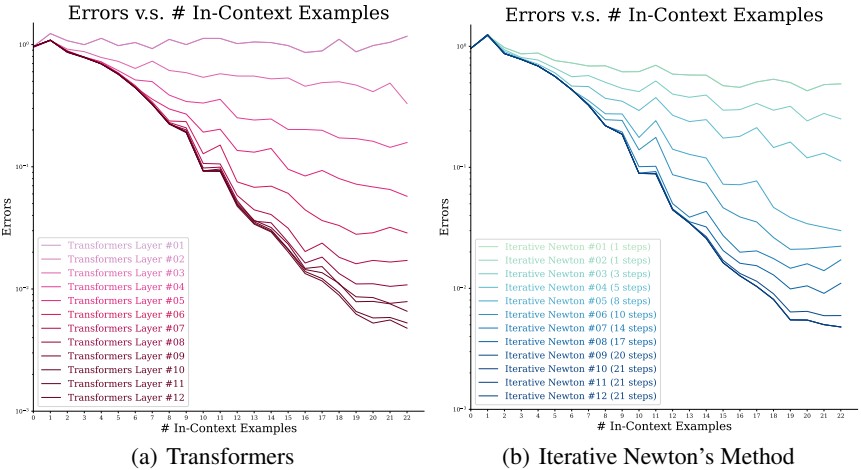

(a) Transformers  (b) Iterative Newton's Method

Figure 15: **Progression of Algorithms on Ill-Conditioned Data.** Transformer's performance still improves over the layer index $\ell$; Iterative Newton's Method's performance improves over the number of iterations $t$ and we plot the best-matching $t$ to Transformer's $\ell$ following Definition 3.4.

We also present the heatmaps to find the best-matching steps and conclude that Transformers are similar to Newton's method than GD in ill-conditioned data.

**Similarity of Errors $\mathcal{E}$ (Transformers v.s. Iterative Newton)**

Trasformer Layer Index

| Iterative newton Steps | 1 | 2 | 3 | 4 | 5 | 6 | 7 | 8 | 9 | 10 | 11 | 12 |
|---|---|---|---|---|---|---|---|---|---|---|---|---|
| 1 | .885 | .886 | .829 | .713 | .598 | .557 | .535 | .529 | .528 | .530 | .532 | .529 |
| 2 | .814 | .814 | .848 | .780 | .662 | .615 | .593 | .587 | .585 | .587 | .589 | .586 |
| 3 | .736 | .736 | .842 | .838 | .733 | .679 | .656 | .650 | .649 | .650 | .652 | .650 |
| 4 | .661 | .662 | .811 | .878 | .805 | .745 | .722 | .716 | .714 | .715 | .716 | .715 |
| 5 | .593 | .593 | .765 | .893 | .867 | .808 | .783 | .777 | .775 | .775 | .777 | .775 |
| 6 | .536 | .537 | .715 | .887 | .913 | .862 | .834 | .828 | .825 | .826 | .827 | .826 |
| 7 | .493 | .494 | .672 | .868 | .940 | .903 | .873 | .866 | .864 | .864 | .865 | .864 |
| 8 | .464 | .464 | .640 | .847 | .951 | .933 | .902 | .894 | .892 | .893 | .894 | .893 |
| 9 | .444 | .445 | .617 | .828 | .953 | .953 | .923 | .915 | .913 | .913 | .914 | .913 |
| 10 | .431 | .432 | .601 | .812 | .948 | .966 | .938 | .930 | .928 | .928 | .929 | .928 |
| 11 | .422 | .423 | .590 | .800 | .942 | .973 | .949 | .940 | .939 | .939 | .939 | .939 |
| 12 | .416 | .416 | .582 | .791 | .935 | .976 | .958 | .949 | .947 | .947 | .948 | .948 |
| 13 | .411 | .412 | .576 | .784 | .928 | .977 | .965 | .956 | .954 | .954 | .955 | .956 |
| 14 | .407 | .408 | .572 | .778 | .923 | .976 | .971 | .963 | .961 | .962 | .962 | .963 |
| 15 | .404 | .404 | .567 | .772 | .916 | .973 | .976 | .970 | .968 | .968 | .969 | .970 |
| 16 | .400 | .400 | .563 | .766 | .910 | .970 | .980 | .975 | .974 | .974 | .975 | .976 |
| 17 | .397 | .397 | .559 | .760 | .904 | .966 | .981 | .979 | .978 | .979 | .979 | .980 |
| 18 | .394 | .394 | .555 | .756 | .898 | .962 | .982 | .983 | .982 | .982 | .983 | .984 |
| 19 | .392 | .392 | .552 | .752 | .894 | .958 | .981 | .985 | .984 | .985 | .986 | .986 |
| 20 | .390 | .390 | .549 | .748 | .890 | .954 | .979 | .985 | .985 | .986 | .987 | .988 |
| 21 | .389 | .389 | .548 | .746 | .887 | .951 | .977 | .985 | .985 | .986 | .987 | .988 |
| 22 | .387 | .388 | .545 | .743 | .883 | .947 | .973 | .983 | .983 | .984 | .985 | .986 |
| 23 | .384 | .385 | .538 | .733 | .872 | .935 | .962 | .972 | .972 | .973 | .974 | .975 |

**Similarity of Errors $\mathcal{E}$ (Transformers v.s. Gradient Descent)**

Trasformer Layer Index

| Gradient descent Steps | 1 | 2 | 3 | 4 | 5 | 6 | 7 | 8 | 9 | 10 | 11 | 12 |
|---|---|---|---|---|---|---|---|---|---|---|---|---|
| 1 | .990 | .990 | .709 | .548 | .469 | .440 | .420 | .413 | .413 | .416 | .418 | .413 |
| 100 | .502 | .503 | .686 | .870 | .941 | .921 | .896 | .889 | .886 | .887 | .887 | .886 |
| 200 | .451 | .451 | .633 | .839 | .953 | .958 | .936 | .929 | .927 | .927 | .927 | .926 |
| 300 | .433 | .433 | .612 | .821 | .950 | .970 | .952 | .945 | .943 | .943 | .943 | .943 |
| 400 | .422 | .423 | .600 | .809 | .945 | .975 | .960 | .954 | .952 | .952 | .952 | .952 |
| 500 | .417 | .418 | .593 | .802 | .941 | .977 | .966 | .960 | .958 | .958 | .958 | .958 |
| 600 | .413 | .413 | .588 | .796 | .937 | .978 | .970 | .964 | .962 | .962 | .962 | .962 |
| 700 | .410 | .410 | .584 | .791 | .933 | .978 | .973 | .967 | .965 | .965 | .966 | .966 |
| 800 | .408 | .408 | .581 | .788 | .930 | .978 | .975 | .970 | .968 | .968 | .968 | .968 |
| 900 | .405 | .406 | .578 | .785 | .927 | .977 | .977 | .972 | .970 | .970 | .970 | .971 |
| 1000 | .404 | .405 | .576 | .782 | .925 | .977 | .978 | .974 | .972 | .972 | .972 | .972 |
| 1100 | .402 | .403 | .574 | .780 | .923 | .976 | .979 | .975 | .974 | .974 | .974 | .974 |
| 1200 | .401 | .402 | .573 | .778 | .921 | .975 | .980 | .976 | .975 | .975 | .975 | .976 |
| 1300 | .400 | .400 | .572 | .776 | .919 | .975 | .981 | .977 | .976 | .976 | .976 | .977 |
| 1400 | .399 | .400 | .571 | .775 | .918 | .974 | .981 | .978 | .977 | .977 | .977 | .978 |
| 1500 | .399 | .400 | .570 | .774 | .917 | .974 | .982 | .980 | .978 | .979 | .979 | .979 |
| 1600 | .398 | .398 | .569 | .772 | .915 | .973 | .982 | .980 | .979 | .979 | .979 | .980 |
| 1700 | .397 | .398 | .568 | .771 | .913 | .972 | .982 | .981 | .979 | .980 | .980 | .980 |
| 1800 | .397 | .397 | .567 | .770 | .913 | .971 | .983 | .982 | .980 | .981 | .981 | .981 |
| 1900 | .396 | .396 | .567 | .769 | .912 | .971 | .983 | .982 | .981 | .981 | .981 | .982 |
| 2000 | .395 | .396 | .566 | .768 | .910 | .970 | .983 | .982 | .981 | .982 | .982 | .982 |
| 2100 | .395 | .395 | .565 | .767 | .909 | .970 | .983 | .983 | .982 | .982 | .982 | .983 |
| 2200 | .394 | .394 | .564 | .766 | .908 | .969 | .983 | .983 | .982 | .982 | .983 | .983 |
| 2300 | .394 | .395 | .564 | .766 | .908 | .969 | .984 | .984 | .982 | .983 | .983 | .984 |
| 2400 | .393 | .393 | .563 | .765 | .907 | .968 | .983 | .984 | .983 | .983 | .983 | .984 |
| 2500 | .393 | .394 | .563 | .765 | .907 | .968 | .984 | .985 | .984 | .984 | .984 | .985 |
| 2600 | .393 | .394 | .563 | .764 | .905 | .967 | .984 | .985 | .984 | .984 | .984 | .985 |
| 2700 | .393 | .394 | .562 | .763 | .905 | .967 | .984 | .985 | .984 | .984 | .984 | .985 |
| 2800 | .392 | .392 | .562 | .763 | .904 | .966 | .983 | .985 | .984 | .984 | .984 | .985 |
| 2900 | .392 | .392 | .561 | .762 | .903 | .965 | .983 | .985 | .984 | .984 | .985 | .985 |
| 3000 | .391 | .392 | .561 | .762 | .903 | .965 | .984 | .985 | .984 | .985 | .985 | .986 |

Figure 16: **Similarity of Errors on Ill-Conditioned Data.** The best matching steps are highlighted in yellow.

**Similarity of Induced Weight $\tilde{w}$ (Transformers v.s. Iterative Newton)**

**Similarity of Induced Weight $\tilde{w}$ (Transformers v.s. Gradient Descent)**

Figure 17: **Similarity of Induced Weights on Ill-Conditioned Data.** The best matching steps are highlighted in yellow.

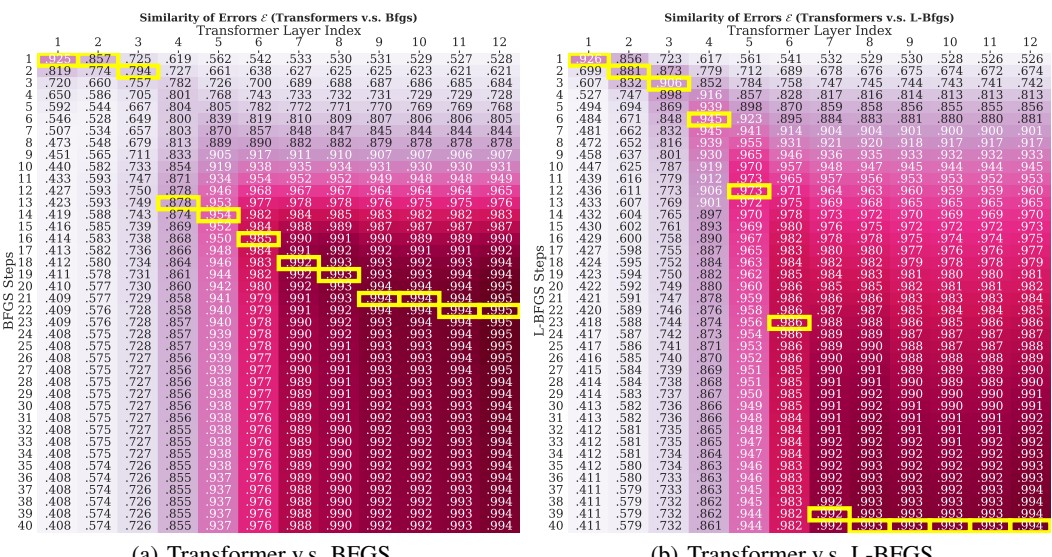

(a) Transformer v.s. BFGS  (b) Transformer v.s. L-BFGS

Figure 18: **Similarity of Errors on Ill-Conditioned Data with Quasi-Newton Methods.** The best matching steps are highlighted in yellow. Transformer also matches BFGS linearly, from layers 4 to 11. L-BFGS still suffers due to its limited memory but still better than Gradient Descent because L-BFGS also attempts to approximate second-order information.

### A.3.2 Experiments with Noisy Linear Regression

We repeat the same experiments on noisy linear regression tasks with $y = \boldsymbol{w}^\top \boldsymbol{x} + \varepsilon$ where $\varepsilon \sim \mathcal{N}(0, \sigma^2)$ with noise level $\sigma = 0.1$. As shown in Figure 19, Transformers still show superlinear convergence on noisy linear regression tasks. Since the predictor is $\hat{\boldsymbol{w}} = \left(\boldsymbol{X}^\top \boldsymbol{X} + \lambda \boldsymbol{I}\right)^\dagger \boldsymbol{X}^\top \boldsymbol{y}$ for some $\lambda$, the iterative newton's method is applied to $\boldsymbol{S} = \boldsymbol{X}^\top \boldsymbol{X} + \lambda \boldsymbol{I}$. Iterative Newton's method still keeps the same superlinear convergence rates. As it's also shown in Figure 19, Transformers and Iternative Newton's rates match linearly, as in the noiseless linear regression tasks.

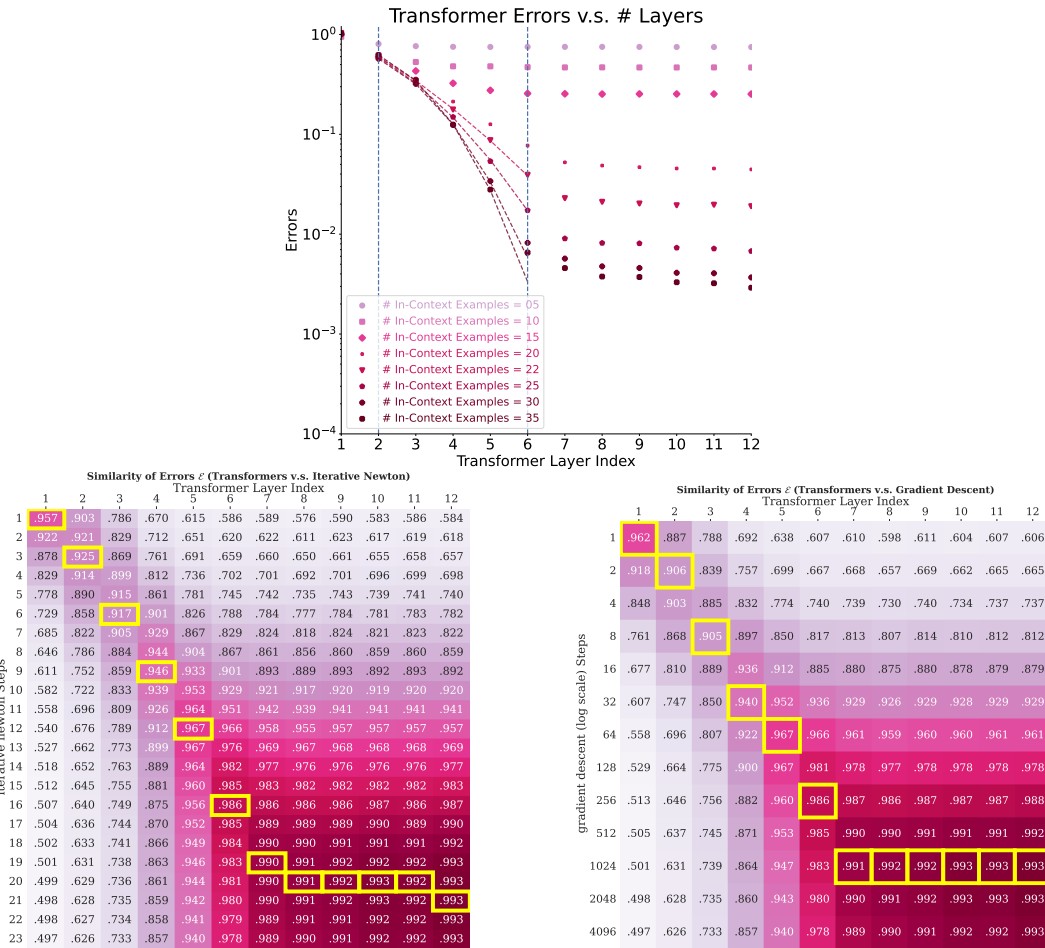

Figure 19: Experiment results on **Noisy Linear Regression**. **(Top)** Transformers have superlinear convergence rate. **(Bottom)** Transformers match Iterative Newton's rate and are exponentially faster than Gradient Descent.

### A.3.3 Experiments with a Non-Linear Function Class (2-Layer MLP)

To extend our experiments to non-linear cases, we adopt the same 2-layer ReLU neural network studied by Garg et al. [2022]: see Fig. 5(c) in their paper. For any prompt $(\boldsymbol{x}_1, y_1, \cdots, \boldsymbol{x}_t, y_t)$, instead of generating labels $y_k = \boldsymbol{w}^{\star \top} \boldsymbol{x}$ as mainly studied in the paper, we study a 2-layer neural network function class parameterized by $\boldsymbol{W} \in \mathbb{R}^{d_{\text{hidden}} \times d}$, $\boldsymbol{v} \in \mathbb{R}^{d_{\text{hidden}}}$, $\boldsymbol{a} \in \mathbb{R}^{d_{\text{hidden}}}$, and $b \in \mathbb{R}$, so that

$$y_k = f_{\boldsymbol{W}, \boldsymbol{v}, \boldsymbol{a}, b}(\boldsymbol{x}_k) = \boldsymbol{a}^\top \text{ReLU}\left(\boldsymbol{W} \boldsymbol{x}_k + \boldsymbol{v}\right) + b \tag{15}$$

Then we repeat the same probing experiments as in the main paper. As shown in Figure 20, even on 2-layer neural network tasks with ReLU activation, Transformer shows superlinear convergence rates. Transformer shows an exponentially faster convergence rate than Gradient Descent's, because

Gradient Descent's steps are shown in log scale and the trend is linear – similar to Figure 9 in the main paper.

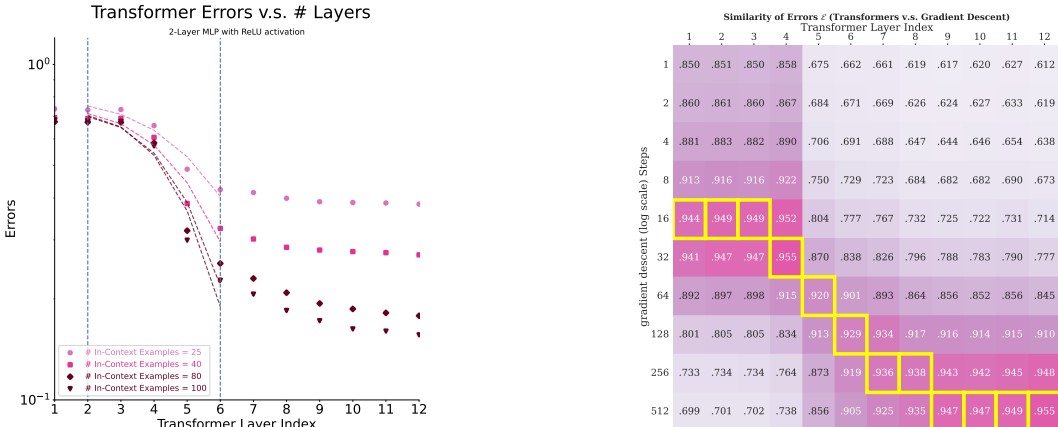

Figure 20: Empirical Results on 2-Layer Neural Network Regression with ReLU activation function. Transformers have superlinear convergence rates and match Gradient Descent's convergence rate exponentially

It would be interesting to ablate the activation function used in Equation (16). We further consider the case when it's using the Tanh activation instead of ReLU, i.e.

$$y_k = f_{\boldsymbol{W}, \boldsymbol{v}, \boldsymbol{a}, b}(\boldsymbol{x}_k) = \boldsymbol{a}^\top \mathrm{Tanh}\Big(\boldsymbol{W}\boldsymbol{x}_k + \boldsymbol{v}\Big) + b \tag{16}$$

Repeating the same experiments as before, as shown in Figure 21, we find that Transformers use the entire first 5 layers to pre-process and then only in the next few layers show exponentially faster convergence rate compared to Gradient Descent. We further note that in both Figure 20 and Figure 21, the cosine similarities between Transformers and Gradient Descent are significantly lower than the experiments with linear regression tasks. This might due to the over-parameterization of the function class and Transformers and Gradient Descent may arrive at different optima.

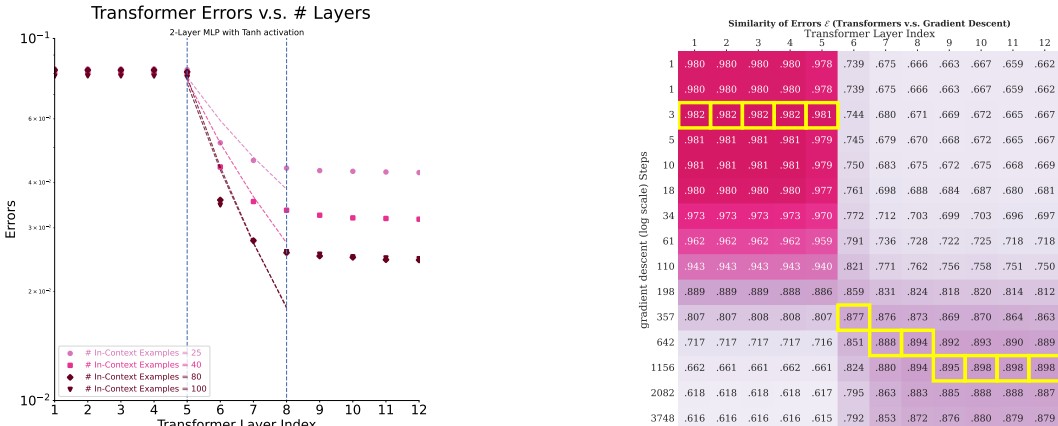

Figure 21: Empirical Results on 2-Layer Neural Network Regression with Tanh activation function. Transformers have superlinear convergence rates and match Gradient Descent's convergence rate exponentially

It would be interesting for future research to explore further this function class of 2-layer MLP to understand fully how Transformer solve the regression problem in-context and whether it achieves a different optimum compared to alternative algorithms such as (Stochastic) Gradient Descent.

## A.4 Varying Transformer Architecture

### A.4.1 Experiments on Transformers of Fewer Heads

In this section, we present experimental results from an alternative model configurations than the main text. We show in the main text that Transformers learn second-order optimization methods in-context where the experiments are using a GPT-2 model with 12 layers and 8 heads per layer. In this section, we present experiments with a GPT-2 model with 12 layers but only 1 head per layer.

**Similarity of Errors $\mathcal{E}$ (Transformers v.s. Iterative Newton)**
Transformer Layer Index

| Iterative newton Steps | 1 | 2 | 3 | 4 | 5 | 6 | 7 | 8 | 9 | 10 | 11 | 12 |
|---|---|---|---|---|---|---|---|---|---|---|---|---|
| 1 | .920 | .920 | .911 | .909 | .861 | .785 | .707 | .671 | .647 | .631 | .626 | .619 |
| 2 | .876 | .876 | .892 | .912 | .879 | .823 | .749 | .709 | .685 | .667 | .663 | .655 |
| 3 | .829 | .829 | .864 | .901 | .887 | .859 | .791 | .750 | .726 | .706 | .702 | .694 |
| 4 | .780 | .780 | .829 | .877 | .884 | .887 | .832 | .792 | .768 | .746 | .743 | .735 |
| 5 | .733 | .733 | .791 | .845 | .872 | .906 | .867 | .832 | .810 | .787 | .784 | .776 |
| 6 | .690 | .690 | .753 | .811 | .853 | .913 | .896 | .869 | .849 | .825 | .823 | .816 |
| 7 | .654 | .654 | .719 | .777 | .829 | .910 | .916 | .900 | .884 | .861 | .860 | .852 |
| 8 | .624 | .624 | .688 | .746 | .805 | .900 | .927 | .924 | .912 | .894 | .893 | .885 |
| 9 | .598 | .598 | .661 | .719 | .780 | .885 | .930 | .939 | .934 | .920 | .920 | .913 |
| 10 | .576 | .576 | .637 | .695 | .757 | .867 | .926 | .947 | .947 | .941 | .942 | .935 |
| 11 | .559 | .559 | .619 | .676 | .738 | .851 | .918 | .948 | .955 | .956 | .957 | .951 |
| 12 | .546 | .546 | .605 | .662 | .723 | .837 | .910 | .947 | .958 | .966 | .968 | .963 |
| 13 | .537 | .537 | .595 | .651 | .712 | .826 | .903 | .944 | .959 | .973 | .976 | .972 |
| 14 | .530 | .530 | .587 | .644 | .704 | .817 | .896 | .940 | .958 | .977 | .981 | .979 |
| 15 | .525 | .525 | .582 | .638 | .698 | .810 | .890 | .936 | .957 | .980 | .984 | .984 |
| 16 | .522 | .522 | .578 | .634 | .693 | .806 | .886 | .933 | .955 | .981 | .986 | .987 |
| 17 | .519 | .519 | .576 | .631 | .690 | .802 | .882 | .930 | .953 | .981 | .987 | .989 |
| 18 | .518 | .517 | .573 | .629 | .688 | .799 | .880 | .928 | .951 | .981 | .987 | .991 |
| 19 | .516 | .516 | .572 | .627 | .686 | .798 | .878 | .926 | .949 | .980 | .986 | .992 |
| 20 | .515 | .515 | .571 | .626 | .685 | .796 | .876 | .924 | .948 | .979 | .986 | .992 |
| 21 | .514 | .514 | .570 | .625 | .684 | .795 | .874 | .923 | .946 | .978 | .985 | .992 |
| 22 | .513 | .513 | .569 | .624 | .683 | .793 | .872 | .920 | .945 | .976 | .983 | .991 |
| 23 | .510 | .510 | .565 | .620 | .679 | .787 | .865 | .914 | .938 | .969 | .976 | .984 |

**Similarity of Errors $\mathcal{E}$ (Transformers v.s. Gradient Descent)**
Transformer Layer Index

| Gradient descent Steps | 1 | 2 | 3 | 4 | 5 | 6 | 7 | 8 | 9 | 10 | 11 | 12 |
|---|---|---|---|---|---|---|---|---|---|---|---|---|
| 1 | .954 | .955 | .915 | .885 | .840 | .757 | .685 | .655 | .630 | .615 | .609 | .604 |
| 50 | .584 | .585 | .645 | .703 | .764 | .870 | .923 | .943 | .945 | .943 | .944 | .939 |
| 100 | .552 | .552 | .610 | .668 | .729 | .841 | .911 | .945 | .955 | .964 | .966 | .962 |
| 150 | .539 | .539 | .596 | .653 | .713 | .826 | .902 | .941 | .956 | .970 | .973 | .971 |
| 200 | .532 | .532 | .588 | .645 | .705 | .819 | .897 | .938 | .955 | .973 | .977 | .975 |
| 250 | .528 | .528 | .583 | .640 | .700 | .813 | .892 | .936 | .954 | .974 | .979 | .978 |
| 300 | .525 | .525 | .581 | .637 | .697 | .810 | .889 | .934 | .953 | .975 | .980 | .979 |
| 350 | .522 | .522 | .578 | .635 | .694 | .807 | .887 | .932 | .952 | .975 | .980 | .980 |
| 400 | .520 | .520 | .576 | .632 | .692 | .804 | .885 | .931 | .951 | .976 | .981 | .982 |
| 450 | .519 | .520 | .575 | .631 | .691 | .803 | .884 | .930 | .950 | .976 | .981 | .983 |
| 500 | .519 | .519 | .574 | .630 | .689 | .801 | .882 | .928 | .950 | .976 | .981 | .983 |
| 550 | .518 | .518 | .573 | .629 | .688 | .801 | .881 | .928 | .949 | .976 | .981 | .983 |
| 600 | .517 | .517 | .572 | .628 | .687 | .799 | .880 | .927 | .948 | .976 | .982 | .984 |
| 650 | .516 | .516 | .572 | .628 | .687 | .799 | .880 | .926 | .948 | .976 | .982 | .985 |
| 700 | .516 | .516 | .571 | .627 | .686 | .798 | .879 | .926 | .948 | .976 | .981 | .985 |
| 750 | .516 | .516 | .570 | .626 | .686 | .797 | .878 | .925 | .947 | .976 | .981 | .985 |
| 800 | .516 | .516 | .571 | .626 | .685 | .797 | .878 | .925 | .947 | .976 | .982 | .985 |
| 850 | .515 | .515 | .570 | .626 | .685 | .796 | .877 | .924 | .947 | .976 | .982 | .985 |
| 900 | .514 | .514 | .569 | .625 | .684 | .795 | .876 | .924 | .946 | .976 | .982 | .985 |
| 950 | .513 | .514 | .568 | .625 | .684 | .795 | .876 | .923 | .946 | .976 | .981 | .986 |
| 1000 | .514 | .514 | .569 | .624 | .683 | .795 | .876 | .923 | .946 | .976 | .982 | .986 |
| 1050 | .513 | .513 | .568 | .624 | .683 | .795 | .875 | .923 | .946 | .976 | .982 | .986 |
| 1100 | .513 | .513 | .568 | .624 | .683 | .794 | .875 | .922 | .945 | .975 | .981 | .986 |
| 1150 | .513 | .513 | .568 | .624 | .683 | .794 | .875 | .922 | .945 | .975 | .981 | .986 |
| 1200 | .513 | .513 | .567 | .623 | .682 | .794 | .874 | .922 | .945 | .975 | .981 | .986 |
| 1250 | .513 | .513 | .567 | .623 | .682 | .794 | .875 | .922 | .945 | .975 | .981 | .986 |
| 1300 | .513 | .513 | .568 | .623 | .682 | .793 | .874 | .921 | .944 | .975 | .981 | .986 |

Figure 22: **Similarity of Errors on an alternative Transformers Configuration.** The best matching steps are highlighted in yellow.

**Similarity of Induced Weight $\hat{w}$ (Transformers v.s. Iterative Newton)**
Transformer Layer Index

| Iterative newton Steps | 1 | 2 | 3 | 4 | 5 | 6 | 7 | 8 | 9 | 10 | 11 | 12 |
|---|---|---|---|---|---|---|---|---|---|---|---|---|
| 1 | .000 | -.003 | .581 | .756 | .740 | .757 | .734 | .717 | .713 | .710 | .711 | .712 |
| 2 | .001 | -.003 | .590 | .767 | .770 | .802 | .782 | .766 | .762 | .759 | .760 | .761 |
| 3 | .002 | -.002 | .588 | .764 | .789 | .840 | .827 | .811 | .807 | .805 | .806 | .807 |
| 4 | .002 | -.002 | .580 | .751 | .797 | .868 | .864 | .850 | .847 | .844 | .845 | .846 |
| 5 | .003 | -.002 | .567 | .734 | .796 | .885 | .891 | .881 | .878 | .876 | .877 | .877 |
| 6 | .003 | -.002 | .553 | .716 | .789 | .893 | .911 | .905 | .903 | .901 | .902 | .902 |
| 7 | .003 | -.001 | .541 | .700 | .780 | .894 | .923 | .922 | .921 | .921 | .922 | .922 |
| 8 | .003 | -.001 | .531 | .686 | .770 | .891 | .930 | .934 | .935 | .936 | .937 | .937 |
| 9 | .003 | -.001 | .522 | .675 | .761 | .886 | .932 | .941 | .944 | .947 | .948 | .948 |
| 10 | .002 | -.000 | .515 | .666 | .753 | .880 | .932 | .945 | .949 | .954 | .955 | .955 |
| 11 | .003 | -.000 | .510 | .660 | .746 | .875 | .930 | .946 | .952 | .959 | .961 | .961 |
| 12 | .003 | .000 | .506 | .655 | .741 | .870 | .928 | .947 | .954 | .963 | .964 | .965 |
| 13 | .003 | .000 | .503 | .652 | .738 | .867 | .926 | .946 | .954 | .965 | .967 | .968 |
| 14 | .003 | .001 | .501 | .650 | .736 | .864 | .924 | .945 | .954 | .966 | .968 | .969 |
| 15 | .003 | .001 | .500 | .648 | .734 | .862 | .922 | .944 | .954 | .967 | .969 | .971 |
| 16 | .003 | .001 | .499 | .647 | .732 | .861 | .921 | .943 | .953 | .968 | .970 | .972 |
| 17 | .003 | .001 | .498 | .646 | .731 | .860 | .920 | .942 | .953 | .968 | .970 | .972 |
| 18 | .003 | .001 | .498 | .645 | .731 | .859 | .919 | .942 | .952 | .968 | .970 | .973 |
| 19 | .003 | .001 | .498 | .645 | .730 | .858 | .919 | .941 | .952 | .968 | .970 | .973 |
| 20 | .003 | .001 | .497 | .644 | .730 | .858 | .918 | .941 | .951 | .967 | .970 | .973 |
| 21 | .003 | .001 | .497 | .644 | .729 | .858 | .918 | .941 | .951 | .967 | .970 | .973 |
| 22 | .003 | .001 | .497 | .644 | .729 | .857 | .918 | .940 | .951 | .967 | .970 | .973 |
| 23 | .003 | .001 | .497 | .644 | .729 | .857 | .918 | .940 | .951 | .967 | .969 | .973 |

**Similarity of Induced Weight $\hat{w}$ (Transformers v.s. Gradient Descent)**
Transformer Layer Index

| Gradient descent Steps | 1 | 2 | 3 | 4 | 5 | 6 | 7 | 8 | 9 | 10 | 11 | 12 |
|---|---|---|---|---|---|---|---|---|---|---|---|---|
| 1 | .001 | .119 | .522 | .684 | .689 | .702 | .688 | .673 | .669 | .668 | .669 | |
| 50 | .003 | .020 | .517 | .675 | .758 | .885 | .936 | .951 | .955 | .961 | .961 | .963 |
| 100 | .002 | .019 | .508 | .662 | .746 | .876 | .933 | .952 | .959 | .968 | .968 | .970 |
| 150 | .002 | .019 | .503 | .657 | .741 | .871 | .930 | .952 | .959 | .970 | .971 | .973 |
| 200 | .003 | .019 | .502 | .655 | .739 | .869 | .928 | .951 | .959 | .971 | .972 | .974 |
| 250 | .003 | .019 | .501 | .653 | .737 | .867 | .927 | .950 | .959 | .971 | .972 | .975 |
| 300 | .002 | .019 | .500 | .652 | .736 | .866 | .927 | .950 | .958 | .971 | .972 | .975 |
| 350 | .003 | .019 | .499 | .652 | .735 | .865 | .926 | .950 | .958 | .972 | .973 | .976 |
| 400 | .003 | .019 | .499 | .651 | .735 | .865 | .925 | .949 | .958 | .972 | .973 | .976 |
| 450 | .003 | .019 | .498 | .651 | .734 | .864 | .925 | .949 | .958 | .972 | .973 | .976 |
| 500 | .003 | .019 | .498 | .650 | .734 | .864 | .925 | .949 | .958 | .972 | .973 | .976 |
| 550 | .003 | .019 | .498 | .650 | .734 | .863 | .924 | .948 | .957 | .972 | .973 | .976 |
| 600 | .003 | .019 | .498 | .650 | .733 | .863 | .924 | .948 | .957 | .972 | .973 | .976 |
| 650 | .002 | .019 | .498 | .650 | .733 | .863 | .924 | .948 | .957 | .972 | .973 | .977 |
| 700 | .003 | .019 | .497 | .649 | .733 | .863 | .924 | .948 | .957 | .972 | .973 | .977 |
| 750 | .003 | .019 | .497 | .649 | .733 | .862 | .923 | .948 | .957 | .972 | .973 | .977 |
| 800 | .003 | .019 | .497 | .649 | .733 | .862 | .923 | .948 | .957 | .972 | .973 | .977 |
| 850 | .003 | .019 | .497 | .649 | .733 | .862 | .923 | .948 | .957 | .972 | .973 | .977 |
| 900 | .003 | .019 | .497 | .649 | .732 | .862 | .923 | .948 | .957 | .972 | .973 | .977 |
| 950 | .003 | .019 | .497 | .649 | .732 | .862 | .923 | .947 | .957 | .972 | .973 | .977 |
| 1000 | .003 | .019 | .497 | .649 | .732 | .862 | .923 | .947 | .957 | .972 | .973 | .977 |
| 1050 | .003 | .019 | .497 | .649 | .732 | .862 | .922 | .947 | .957 | .972 | .973 | .977 |
| 1100 | .003 | .019 | .497 | .649 | .732 | .862 | .923 | .947 | .957 | .972 | .973 | .977 |
| 1150 | .003 | .019 | .497 | .649 | .732 | .862 | .923 | .947 | .956 | .972 | .973 | .977 |
| 1200 | .003 | .019 | .496 | .648 | .732 | .861 | .922 | .947 | .956 | .972 | .973 | .977 |
| 1250 | .003 | .019 | .497 | .648 | .732 | .861 | .922 | .947 | .956 | .972 | .973 | .977 |
| 1300 | .003 | .019 | .496 | .648 | .732 | .861 | .922 | .947 | .956 | .972 | .973 | .977 |

Figure 23: **Similarity of Induced Weights on an alternative Transformers Configuration.** The best matching steps are highlighted in yellow.

We conclude that our experimental results are not restricted to a specific model configurations, smaller models such as GPT-2 with 12 layers and 1 head each layer also suffice in implementing the Iterative Newton's method, and more similar than gradient descents, in terms of rate of convergence.

### A.4.2 Experiments on Transformers with More Layers

In this section, we investigate whether deeper models would behave similarly or differently. We work on Transformers with 24 layers and 8 heads each.

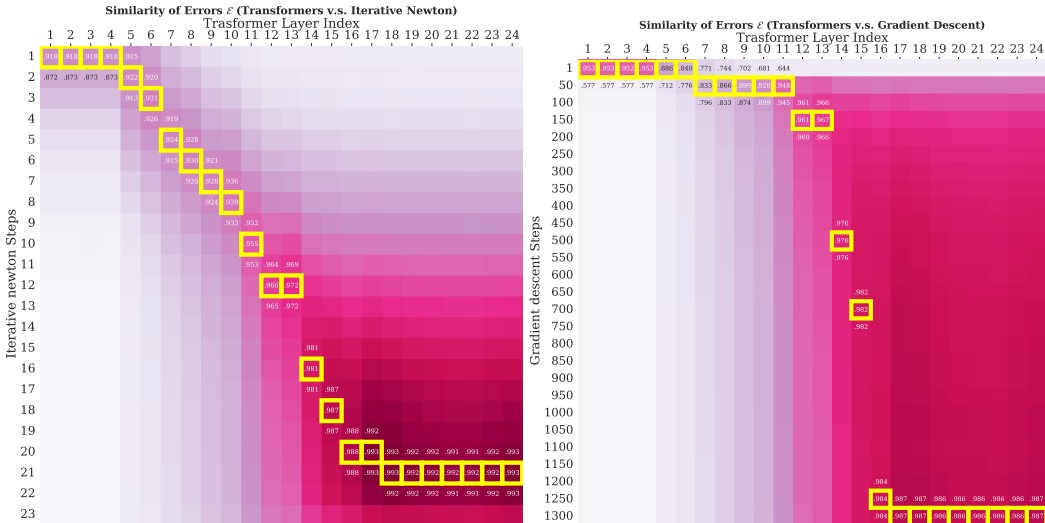

Figure 24: **Similarity of Errors on a 24-layer Transformers Configuration.** The best matching steps are highlighted in yellow.

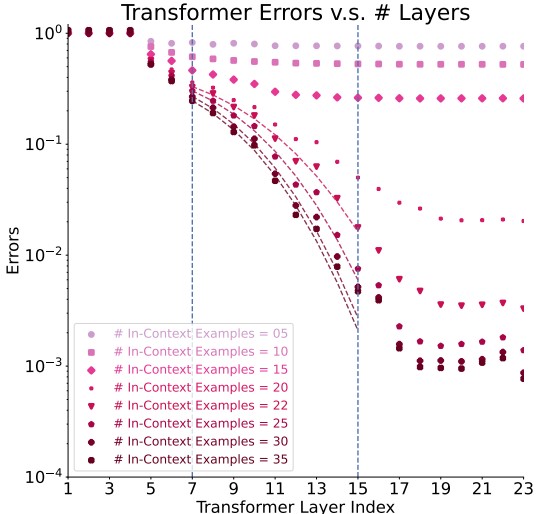

Figure 25: Transformers with 24 layers also converge superlinearly, similar to Iterative Newton.

## A.5 Heatmaps with Best-Matching Steps Help Compare Convergence Rates

In this section, we show the heatmaps with best-matching steps among *known algorithms.*

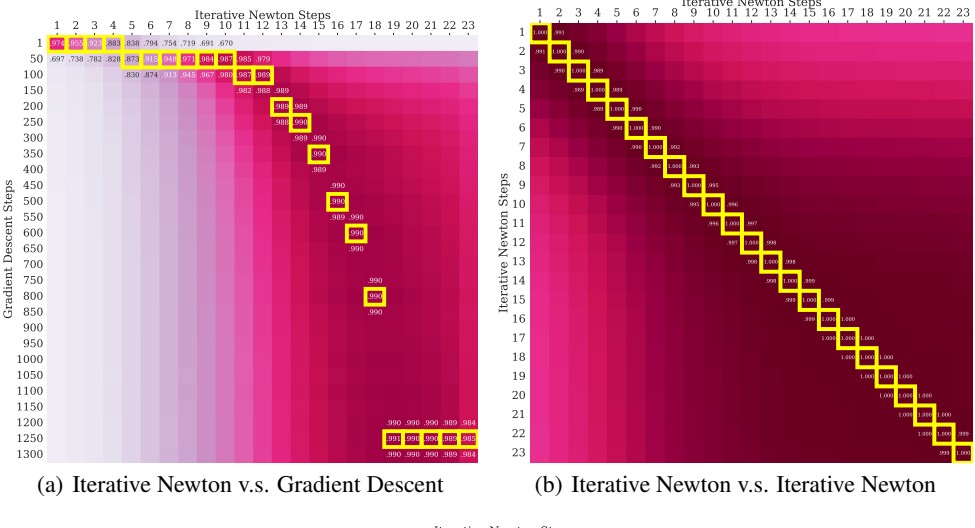

(a) Iterative Newton v.s. Gradient Descent      (b) Iterative Newton v.s. Iterative Newton

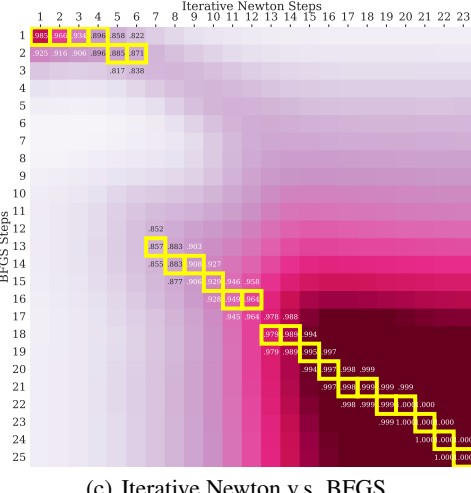

(c) Iterative Newton v.s. BFGS

Figure 26: Best-Matching Steps on Similarity of Residuals Help Compare Convergence Rates. (a: top-left) When comparing Iterative Newton and Gradient Descent, there is an exponential trend – showing Iterative Newton converges exponentially faster than Gradient Descent. (b: top-right) When Iterative Newton is compared with itself in sub-figure, there is a linear trend – showing they have the same convergence rate. (c: bottom) When Iterative Newton is compared to BFGS in sub-figure, there a linear trend after there are enough steps for BFGS to approximate second-order information – showing Iterative Newton and BFGS share a similar convergence rate after sufficient BFGS steps.

## A.6 Definitions for Evaluating Forgetting

We measure the phenomenon of model forgetting by reusing an in-context example within $\{\boldsymbol{x}_i, y_i\}_{i=1}^n$ as the test example $\boldsymbol{x}_{\text{test}}$. In experiments of Figure 5, we fix $n = 20$ and reuse $\boldsymbol{x}_{\text{test}} = \boldsymbol{x}_i$. We denote the "Time Stamp Gap" as the distance the reused example index $i$ from the current time stamp $n = 20$. We measure the forgetting of index $i$ as

$$\text{Forgetting}(\mathcal{A}, i) = \underset{\{\boldsymbol{x}_i, y_i\}_{i=1}^n \sim P_{\mathcal{D}}}{\mathbb{E}} \text{MSE}\Big(\mathcal{A}(\boldsymbol{x}_i \mid \{\boldsymbol{x}_i, y_i\}_{i=1}^n), y_i\Big) \tag{17}$$

Note: the further away $i$ is from $n$, the more possible algorithm $\mathcal{A}$ forgets.

# B Detailed Proofs for Section 5

In this section, we work on full attention layers with normalized ReLU activation $\sigma(\cdot) = \frac{1}{n}\text{ReLU}(\cdot)$ given $n$ examples.

**Definition B.1.** A full attention layer with $M$ heads and ReLU activation is also denoted as $\text{Attn}$ on any input sequence $\boldsymbol{H} = [\boldsymbol{h}_1, \cdots, \boldsymbol{h}_N] \in \mathbb{R}^{D \times N}$, where $D$ is the dimension of hidden states and $N$ is the sequence length. In the vector form,

$$\tilde{\boldsymbol{h}}_t = [\text{Attn}(\boldsymbol{H})]_t = \boldsymbol{h}_t + \frac{1}{n} \sum_{m=1}^{M} \sum_{j=1}^{n} \text{ReLU}\left(\langle \boldsymbol{Q}_m \boldsymbol{h}_t, \boldsymbol{K}_m \boldsymbol{h}_j \rangle\right) \cdot \boldsymbol{V}_m \boldsymbol{h}_j \tag{18}$$

*Remark* B.2. This is slightly different from the **causal** attention layer (see Definition 3.1) in that at each time stamp $t$, the attention layer in Definition B.1 has full information of all hidden states $j \in [n]$, unlike causal attention layer which requires $j \in [t]$.

## B.1 Helper Results

We begin by constructing a useful component for our proof, and state some existing constructions from Akyürek et al. [2022].

**Lemma B.3.** *Given hidden states $\{\boldsymbol{h}_1, \cdots, \boldsymbol{h}_n\}$, there exists query, key and value matrices $\boldsymbol{Q}, \boldsymbol{K}, \boldsymbol{V}$ respectively such that one attention layer can compute $\sum_{j=1}^{n} \boldsymbol{h}_j$.*

*Proof.* We can pad each hidden state by 1 and 0's such that $\boldsymbol{h}'_t \leftarrow \begin{bmatrix} \boldsymbol{h}_t \\ 1 \\ \boldsymbol{0}_d \end{bmatrix} \in \mathbb{R}^{2d+1}$. We construct two heads where $\boldsymbol{Q}_1 = \boldsymbol{K}_1 = \boldsymbol{Q}_2 = \begin{bmatrix} \boldsymbol{O}_{d \times d} & \boldsymbol{O}_{d \times 1} & \boldsymbol{O}_{d \times d} \\ \boldsymbol{O}_{1 \times d} & 1 & \boldsymbol{O}_{1 \times d} \\ \boldsymbol{O}_{d \times d} & \boldsymbol{O}_{d \times 1} & \boldsymbol{O}_{d \times d} \end{bmatrix}$ and $\boldsymbol{K}_2 = -\boldsymbol{K}_1$. Then $\begin{bmatrix} \boldsymbol{O}_{d \times d} & \boldsymbol{O}_{d \times 1} & \boldsymbol{O}_{d \times d} \\ \boldsymbol{O}_{1 \times d} & 1 & \boldsymbol{O}_{1 \times d} \\ \boldsymbol{O}_{d \times d} & \boldsymbol{O}_{d \times 1} & \boldsymbol{O}_{d \times d} \end{bmatrix} \boldsymbol{h}'_t = \begin{bmatrix} \boldsymbol{0}_d \\ 1 \\ \boldsymbol{0}_d \end{bmatrix}$.

Let $\boldsymbol{V}_1 = \boldsymbol{V}_2 = \begin{bmatrix} \boldsymbol{O}_{(d+1) \times d} & \boldsymbol{O}_{(d+1) \times (d+1)} \\ n\boldsymbol{I}_{d \times d} & \boldsymbol{O}_{d \times (d+1)} \end{bmatrix}$ so that $\boldsymbol{V}_m \begin{bmatrix} \boldsymbol{h}_j \\ 1 \\ \boldsymbol{0}_d \end{bmatrix} = \begin{bmatrix} \boldsymbol{0}_{d+1} \\ n\boldsymbol{h}_j \end{bmatrix}$.

We apply one attention layer to these 1-padded hidden states and we have

$$\tilde{\boldsymbol{h}}_t = \boldsymbol{h}'_t + \frac{1}{n} \sum_{m=1}^{2} \sum_{j=1}^{n} \text{ReLU}\left(\langle \boldsymbol{Q}_m \boldsymbol{h}'_t, \boldsymbol{K}_m \boldsymbol{h}'_j \rangle\right) \cdot \boldsymbol{V}_m \boldsymbol{h}'_j$$

$$= \boldsymbol{h}'_t + \frac{1}{n} \sum_{j=1}^{n} \left[\text{ReLU}(1) + \text{ReLU}(-1)\right] \cdot \begin{bmatrix} \boldsymbol{0}_{d+1} \\ n\boldsymbol{h}_j \end{bmatrix} \tag{19}$$

$$= \begin{bmatrix} \boldsymbol{h}_t \\ 1 \\ \boldsymbol{0}_d \end{bmatrix} + \begin{bmatrix} \boldsymbol{0}_{d+1} \\ \sum_{j=1}^{n} \boldsymbol{h}_j \end{bmatrix} = \begin{bmatrix} \boldsymbol{h}_t \\ 1 \\ \sum_{j=1}^{n} \boldsymbol{h}_j \end{bmatrix}$$

$\square$

**Proposition B.4** (Akyürek et al., 2022). *Each of $\mathtt{mov}$, $\mathtt{aff}$, $\mathtt{mul}$, $\mathtt{div}$ can be implemented by a single transformer layer. These four operations are mappings $\mathbb{R}^{D \times N} \to \mathbb{R}^{D \times N}$, expressed as follows,*

$\mathtt{mov}(\boldsymbol{H}; s, t, i, j, i', j')$: *selects the entries of the $s$-th column of $\boldsymbol{H}$ between rows $i$ and $j$, and copies them into the $t$-th column ($t \geq s$) of $\boldsymbol{H}$ between rows $i'$ and $j'$.*

$\mathtt{mul}(\boldsymbol{H}; a, b, c, (i, j), (i', j'), (i'', j''))$: *in each column $\boldsymbol{h}$ of $\boldsymbol{H}$, interprets the entries between $i$ and $j$ as an $a \times b$ matrix $\boldsymbol{A}_1$, and the entries between $i'$ and $j'$ as a $b \times c$ matrix $\boldsymbol{A}_2$, multiplies these matrices together, and stores the result between rows $i''$ and $j''$, yielding a matrix in which each column has the form $[\boldsymbol{h}_{:i''-1}, \boldsymbol{A}_1 \boldsymbol{A}_2, \boldsymbol{h}_{j'':}]^\top$. This allows the layer to implement inner products.*

`div`$(\boldsymbol{H};(i,j),i',(i'',j''))$: *in each column $\boldsymbol{h}$ of $\boldsymbol{H}$, divides the entries between $i$ and $j$ by the absolute value of the entry at $i'$, and stores the result between rows $i''$ and $j''$, yielding a matrix in which every column has the form $[\boldsymbol{h}_{:i''-1}, \boldsymbol{h}_{i:j}/|\boldsymbol{h}_{i'}|, \boldsymbol{h}_{j'':}]^\top$.*

`aff`$(\boldsymbol{H};(i,j),(i',j'),(i'',j''),\boldsymbol{W}_1,\boldsymbol{W}_2,\mathbf{b})$: *in each column $\boldsymbol{h}$ of $\boldsymbol{H}$, applies an affine transformation to the entries between $i$ and $j$ and $i'$ and $j'$, then stores the result between rows $i''$ and $j''$, yielding a matrix in which every column has the form $[\boldsymbol{h}_{:i''-1}, \boldsymbol{W}_1\boldsymbol{h}_{i:j} + \boldsymbol{W}_2\boldsymbol{h}_{i':j'} + \mathbf{b}, \boldsymbol{h}_{j'':}]^\top$. This allows the layer to implement subtraction by setting $\boldsymbol{W}_1 = \boldsymbol{I}$ and $\boldsymbol{W}_2 = -\boldsymbol{I}$.*

### B.2 Proof of Theorem 5.1

**Theorem 5.1.** *For any $k$, there exist Transformer weights such that on any set of in-context examples $\{\boldsymbol{x}_i, y_i\}_{i=1}^n$ and test point $\boldsymbol{x}_{\text{test}}$, the Transformer predicts on $\boldsymbol{x}_{\text{test}}$ using $\boldsymbol{x}_{\text{test}}^\top \hat{\boldsymbol{w}}_k^{\text{Newton}}$. Here $\hat{\boldsymbol{w}}_k^{\text{Newton}}$ are the Iterative Newton updates given by $\hat{\boldsymbol{w}}_k^{\text{Newton}} = \boldsymbol{M}_k \boldsymbol{X}^\top \boldsymbol{y}$ where $\boldsymbol{M}_j$ is updated as*

$$\boldsymbol{M}_j = 2\boldsymbol{M}_{j-1} - \boldsymbol{M}_{j-1}\boldsymbol{S}\boldsymbol{M}_{j-1}, 1 \le j \le k, \quad \boldsymbol{M}_0 = \alpha\boldsymbol{S},$$

*for some $\alpha > 0$ and $\boldsymbol{S} = \boldsymbol{X}^\top \boldsymbol{X}$. The dimensionality of the hidden layers is $\mathcal{O}(d)$, and the number of layers is $k + 8$. One transformer layer computes one Newton iteration. 3 initial transformer layers are needed for initializing $\boldsymbol{M}_0$ and 5 layers at the end are needed to read out predictions from the computed pseudo-inverse $\boldsymbol{M}_k$.*

*Proof.* We break the proof into parts.

**Transformers Implement Initialization $\boldsymbol{T}^{(0)} = \alpha\boldsymbol{S}$.** Given input sequence $\boldsymbol{H} := \{\boldsymbol{x}_1, \cdots, \boldsymbol{x}_n\}$, with $\boldsymbol{x}_i \in \mathbb{R}^d$, we first apply the `mov` operations given by Proposition B.4 (similar to Akyürek et al. [2022], we show only non-zero rows when applying these operations):

$$\begin{bmatrix} \boldsymbol{x}_1 & \cdots & \boldsymbol{x}_n \end{bmatrix} \xrightarrow{\text{mov}} \begin{bmatrix} \boldsymbol{x}_1 & \cdots & \boldsymbol{x}_n \\ \boldsymbol{x}_1 & \cdots & \boldsymbol{x}_n \end{bmatrix} \tag{20}$$

We call each column after `mov` as $\boldsymbol{h}_j$. With an full attention layer, one can construct two heads with query and value matrices of the form $\boldsymbol{Q}_1^\top \boldsymbol{K}_1 = -\boldsymbol{Q}_2^\top \boldsymbol{K}_2 = \begin{bmatrix} \boldsymbol{I}_{d\times d} & \boldsymbol{O}_{d\times d} \\ \boldsymbol{O}_{d\times d} & \boldsymbol{O}_{d\times d} \end{bmatrix}$ such that for any $t \in [n]$, we have

$$\sum_{m=1}^2 \text{ReLU}\left(\langle \boldsymbol{Q}_m \boldsymbol{h}_t, \boldsymbol{K}_m \boldsymbol{h}_j \rangle\right) = \text{ReLU}(\boldsymbol{x}_t^\top \boldsymbol{x}_j) + \text{ReLU}(-\boldsymbol{x}_t^\top \boldsymbol{x}_j) = \langle \boldsymbol{x}_t, \boldsymbol{x}_j \rangle \tag{21}$$

Let all value matrices $\boldsymbol{V}_m = n\alpha \begin{bmatrix} \boldsymbol{I}_{d\times d} & \boldsymbol{O}_{d\times d} \\ \boldsymbol{O}_{d\times d} & \boldsymbol{O}_{d\times d} \end{bmatrix}$ for some $\alpha \in \mathbb{R}$. Combining the skip connections, we have

$$\tilde{\boldsymbol{h}}_t = \begin{bmatrix} \boldsymbol{x}_t \\ \boldsymbol{x}_t \end{bmatrix} + \frac{1}{n}\sum_{j=1}^n \langle \boldsymbol{x}_t, \boldsymbol{x}_j \rangle n\alpha \begin{bmatrix} \boldsymbol{x}_j \\ \boldsymbol{0} \end{bmatrix} = \begin{bmatrix} \boldsymbol{x}_t \\ \boldsymbol{x}_t \end{bmatrix} + \begin{bmatrix} \alpha\left(\sum_{j=1}^n \boldsymbol{x}_j \boldsymbol{x}_j^\top\right)\boldsymbol{x}_t \\ \boldsymbol{0} \end{bmatrix} = \begin{bmatrix} \boldsymbol{x}_t + \alpha\boldsymbol{S}\boldsymbol{x}_t \\ \boldsymbol{x}_t \end{bmatrix} \tag{22}$$

Now we can use the `aff` operator to make subtractions and then

$$\begin{bmatrix} \boldsymbol{x}_t + \alpha\boldsymbol{S}\boldsymbol{x}_t \\ \boldsymbol{x}_t \end{bmatrix} \xrightarrow{\text{aff}} \begin{bmatrix} (\boldsymbol{x}_t + \alpha\boldsymbol{S}\boldsymbol{x}_t) - \boldsymbol{x}_t \\ \boldsymbol{x}_t \end{bmatrix} = \begin{bmatrix} \alpha\boldsymbol{S}\boldsymbol{x}_t \\ \boldsymbol{x}_t \end{bmatrix} \tag{23}$$

We call this transformed hidden states as $\boldsymbol{H}^{(0)}$ and denote $\boldsymbol{T}^{(0)} = \alpha\boldsymbol{S}$:

$$\boldsymbol{H}^{(0)} = \begin{bmatrix} \boldsymbol{h}_1^{(0)} & \cdots & \boldsymbol{h}_n^{(0)} \end{bmatrix} = \begin{bmatrix} \boldsymbol{T}^{(0)}\boldsymbol{x}_1 & \cdots & \boldsymbol{T}^{(0)}\boldsymbol{x}_n \\ \boldsymbol{x}_1 & \cdots & \boldsymbol{x}_n \end{bmatrix} \tag{24}$$

Notice that $\boldsymbol{S}$ is symmetric and thereafter $\boldsymbol{T}^{(0)}$ is also symmetric.

**Transformers implement Newton Iteration.** Let the input prompt be the same as Equation (24),

$$\boldsymbol{H}^{(0)} = \begin{bmatrix} \boldsymbol{h}_1^{(0)} & \cdots & \boldsymbol{h}_n^{(0)} \end{bmatrix} = \begin{bmatrix} \boldsymbol{T}^{(0)}\boldsymbol{x}_1 & \cdots & \boldsymbol{T}^{(0)}\boldsymbol{x}_n \\ \boldsymbol{x}_1 & \cdots & \boldsymbol{x}_n \end{bmatrix} \tag{25}$$

We claim that the $\ell$'s hidden states can be of the similar form

$$\boldsymbol{H}^{(\ell)} = \begin{bmatrix} \boldsymbol{h}_1^{(\ell)} & \cdots & \boldsymbol{h}_n^{(\ell)} \end{bmatrix} = \begin{bmatrix} \boldsymbol{T}^{(\ell)}\boldsymbol{x}_1 & \cdots & \boldsymbol{T}^{(\ell)}\boldsymbol{x}_n \\ \boldsymbol{x}_1 & \cdots & \boldsymbol{x}_n \end{bmatrix} \tag{26}$$

We prove by induction that assuming our claim is true for $\ell$, we work on $\ell + 1$:

Let $\boldsymbol{Q}_m = \tilde{\boldsymbol{Q}}_m \underbrace{\begin{bmatrix} \boldsymbol{O}_d & -\frac{n}{2}\boldsymbol{I}_d \\ \boldsymbol{O}_d & \boldsymbol{O}_d \end{bmatrix}}_{\boldsymbol{G}}, \boldsymbol{K}_m = \tilde{\boldsymbol{K}}_m \underbrace{\begin{bmatrix} \boldsymbol{I}_d & \boldsymbol{O}_d \\ \boldsymbol{O}_d & \boldsymbol{O}_d \end{bmatrix}}_{\boldsymbol{J}}$ where $\tilde{\boldsymbol{Q}}_1^\top \tilde{\boldsymbol{K}}_1 := \boldsymbol{I}, \tilde{\boldsymbol{Q}}_2^\top \tilde{\boldsymbol{K}}_2 := -\boldsymbol{I}$ and

$\boldsymbol{V}_1 = \boldsymbol{V}_2 = \underbrace{\begin{bmatrix} \boldsymbol{I}_d & \boldsymbol{O}_d \\ \boldsymbol{O}_d & \boldsymbol{O}_d \end{bmatrix}}_{\boldsymbol{J}}$. A 2-head self-attention layer, with ReLU attentions, can be written has

$$\boldsymbol{h}_t^{(\ell+1)} = [\mathrm{Attn}(\boldsymbol{H}^{(\ell)})]_t = \boldsymbol{h}_t^{(\ell)} + \frac{1}{n}\sum_{m=1}^2 \sum_{j=1}^n \mathrm{ReLU}\left(\left\langle \boldsymbol{Q}_m \boldsymbol{h}_t^{(\ell)}, \boldsymbol{K}_m \boldsymbol{h}_j^{(\ell)} \right\rangle\right) \cdot \boldsymbol{V}_m \boldsymbol{h}_j^{(\ell)} \tag{27}$$

where

$$
\begin{aligned}
&\sum_{m=1}^2 \mathrm{ReLU}\left(\left\langle \boldsymbol{Q}_m \boldsymbol{h}_t^{(\ell)}, \boldsymbol{K}_m \boldsymbol{h}_j^{(\ell)} \right\rangle\right) \cdot \boldsymbol{V}_m \boldsymbol{h}_j^{(\ell)} \\
&= \left[ \mathrm{ReLU}\left((\boldsymbol{G}\boldsymbol{h}_t^{(\ell)})^\top \underbrace{\tilde{\boldsymbol{Q}}_1^\top \tilde{\boldsymbol{K}}_1}_{\boldsymbol{I}} (\boldsymbol{J}\boldsymbol{h}_j^{(\ell)})\right) + \mathrm{ReLU}\left((\boldsymbol{G}\boldsymbol{h}_t^{(\ell)})^\top \underbrace{\tilde{\boldsymbol{Q}}_2^\top \tilde{\boldsymbol{K}}_2}_{-\boldsymbol{I}} (\boldsymbol{J}\boldsymbol{h}_j^{(\ell)})\right) \right] \cdot (\boldsymbol{J}\boldsymbol{h}_j^{(\ell)}) \\
&= \left[ \mathrm{ReLU}((\boldsymbol{G}\boldsymbol{h}_t^{(\ell)})^\top (\boldsymbol{J}\boldsymbol{h}_j^{(\ell)})) + \mathrm{ReLU}(-(\boldsymbol{G}\boldsymbol{h}_t^{(\ell)})^\top (\boldsymbol{J}\boldsymbol{h}_j^{(\ell)})) \right] \cdot (\boldsymbol{J}\boldsymbol{h}_j^{(\ell)}) \\
&= (\boldsymbol{G}\boldsymbol{h}_t^{(\ell)})^\top (\boldsymbol{J}\boldsymbol{h}_j^{(\ell)}) (\boldsymbol{J}\boldsymbol{h}_j^{(\ell)}) \\
&= (\boldsymbol{J}\boldsymbol{h}_j^{(\ell)}) (\boldsymbol{J}\boldsymbol{h}_j^{(\ell)})^\top (\boldsymbol{G}\boldsymbol{h}_t^{(\ell)})
\end{aligned}
\tag{28}
$$

Plug in our assumptions that $\boldsymbol{h}_j^{(\ell)} = \begin{bmatrix} \boldsymbol{T}^{(\ell)}\boldsymbol{x}_j \\ \boldsymbol{x}_j \end{bmatrix}$, we have $\boldsymbol{J}\boldsymbol{h}_j^{(\ell)} = \begin{bmatrix} \boldsymbol{T}^{(\ell)}\boldsymbol{x}_j \\ \boldsymbol{0}_d \end{bmatrix}$ and $\boldsymbol{G}\boldsymbol{h}_t^{(\ell)} = \begin{bmatrix} -\frac{n}{2}\boldsymbol{x}_t \\ \boldsymbol{0}_d \end{bmatrix}$,
we have

$$
\begin{aligned}
\boldsymbol{h}_t^{(\ell+1)} &= \begin{bmatrix} \boldsymbol{T}^{(\ell)}\boldsymbol{x}_t \\ \boldsymbol{x}_t \end{bmatrix} + \frac{1}{n}\sum_{j=1}^n \begin{bmatrix} \boldsymbol{T}^{(\ell)}\boldsymbol{x}_j \\ \boldsymbol{0}_d \end{bmatrix} \begin{bmatrix} \boldsymbol{T}^{(\ell)}\boldsymbol{x}_j \\ \boldsymbol{0}_d \end{bmatrix}^\top \begin{bmatrix} -\frac{n}{2}\boldsymbol{x}_t \\ \boldsymbol{0}_d \end{bmatrix} \\
&= \begin{bmatrix} \boldsymbol{T}^{(\ell)}\boldsymbol{x}_t - \frac{1}{2}\sum_{j=1}^n (\boldsymbol{T}^{(\ell)}\boldsymbol{x}_j)(\boldsymbol{T}^{(\ell)}\boldsymbol{x}_j)^\top \boldsymbol{x}_t \\ \boldsymbol{x}_t \end{bmatrix} \\
&= \begin{bmatrix} \boldsymbol{T}^{(\ell)}\boldsymbol{x}_t - \frac{1}{2}\boldsymbol{T}^{(\ell)}\left(\sum_{j=1}^n \boldsymbol{x}_j\boldsymbol{x}_j^\top\right)\boldsymbol{T}^{(\ell)\top}\boldsymbol{x}_t \\ \boldsymbol{x}_t \end{bmatrix} \\
&= \begin{bmatrix} \left(\boldsymbol{T}^{(\ell)} - \frac{1}{2}\boldsymbol{T}^{(\ell)}\boldsymbol{S}\boldsymbol{T}^{(\ell)\top}\right)\boldsymbol{x}_t \\ \boldsymbol{x}_t \end{bmatrix}
\end{aligned}
\tag{29}
$$

Now we pass over an MLP layer with

$$\boldsymbol{h}_t^{(\ell+1)} \leftarrow \boldsymbol{h}_t^{(\ell+1)} + \begin{bmatrix} \boldsymbol{I}_d & \boldsymbol{O}_d \\ \boldsymbol{O}_d & \boldsymbol{O}_d \end{bmatrix} \boldsymbol{h}_t^{(\ell+1)} = \begin{bmatrix} \left(2\boldsymbol{T}^{(\ell)} - \boldsymbol{T}^{(\ell)}\boldsymbol{S}\boldsymbol{T}^{(\ell)\top}\right)\boldsymbol{x}_t \\ \boldsymbol{x}_t \end{bmatrix} \tag{30}$$

Now we denote the iteration

$$\boldsymbol{T}^{(\ell+1)} = 2\boldsymbol{T}^{(\ell)} - \boldsymbol{T}^{(\ell)}\boldsymbol{S}\boldsymbol{T}^{(\ell)\top} \tag{31}$$

We find that $\boldsymbol{T}^{(\ell+1)\top} = \boldsymbol{T}^{(\ell+1)}$ since $\boldsymbol{T}^{(\ell)}$ and $\boldsymbol{S}$ are both symmetric. It reduces to

$$\boldsymbol{T}^{(\ell+1)} = 2\boldsymbol{T}^{(\ell)} - \boldsymbol{T}^{(\ell)}\boldsymbol{S}\boldsymbol{T}^{(\ell)} \tag{32}$$

This is exactly the same as the Newton iteration.

**Transformers can implement** $\hat{w}_\ell^{\mathrm{TF}} = \boldsymbol{T}^{(\ell)}\boldsymbol{X}^\top \boldsymbol{y}$**.** Going back to the empirical prompt format $\{\boldsymbol{x}_1, y_1, \cdots, \boldsymbol{x}_n, y_n\}$. We can let parameters be zero for positions of $y$'s and only rely on the skip

connection up to layer $\ell$, and the $\boldsymbol{H}^{(\ell)}$ is then $\begin{bmatrix} \boldsymbol{T}^{(\ell)}\boldsymbol{x}_j & \boldsymbol{0} \\ \boldsymbol{x}_j & \boldsymbol{0} \\ 0 & y_j \end{bmatrix}_{j=1}^{n}$ . We again apply operations from

Proposition B.4:

$$\begin{bmatrix} \boldsymbol{T}^{(\ell)}\boldsymbol{x}_j & \boldsymbol{0} \\ \boldsymbol{x}_j & \boldsymbol{0} \\ 0 & y_j \end{bmatrix}_{j=1}^{n} \xrightarrow{\text{mov}} \begin{bmatrix} \boldsymbol{T}^{(\ell)}\boldsymbol{x}_j & \boldsymbol{T}^{(\ell)}\boldsymbol{x}_j \\ \boldsymbol{x}_j & \boldsymbol{0} \\ 0 & y_j \end{bmatrix}_{j=1}^{n} \xrightarrow{\text{mul}} \begin{bmatrix} \boldsymbol{T}^{(\ell)}\boldsymbol{x}_j & \boldsymbol{T}^{(\ell)}\boldsymbol{x}_j \\ \boldsymbol{x}_j & \boldsymbol{0} \\ 0 & y_j \\ \boldsymbol{0} & \boldsymbol{T}^{(\ell)}y_j\boldsymbol{x}_j \end{bmatrix}_{j=1}^{n} \tag{33}$$

Now we apply Lemma B.3 over all even columns in Equation (33) and we have

$$\text{Output} = \sum_{j=1}^{n} \begin{bmatrix} \boldsymbol{T}^{(\ell)}\boldsymbol{x}_j \\ \boldsymbol{0} \\ y_j \\ \boldsymbol{T}^{(\ell)}y_j\boldsymbol{x}_j \end{bmatrix} = \begin{bmatrix} \boldsymbol{\xi} \\ \boldsymbol{T}^{(\ell)}\sum_{j=1}^{n} y_j\boldsymbol{x}_j \end{bmatrix} = \begin{bmatrix} \boldsymbol{\xi} \\ \boldsymbol{T}^{(\ell)}\boldsymbol{X}^{\top}\boldsymbol{y} \end{bmatrix} \tag{34}$$

where $\boldsymbol{\xi}$ denotes irrelevant quantities. Note that the resulting $\boldsymbol{T}^{(\ell)}\boldsymbol{X}^{\top}\boldsymbol{y}$ is also the same as Iterative Newton's predictor $\hat{\boldsymbol{w}}_k = \boldsymbol{M}_k\boldsymbol{X}^{\top}\boldsymbol{y}$ after $k$ iterations. We denote $\hat{\boldsymbol{w}}_\ell^{\text{TF}} = \boldsymbol{T}^{(\ell)}\boldsymbol{X}^{\top}\boldsymbol{y}$.

**Transformers can make predictions on $\boldsymbol{x}_{test}$ by $\langle \hat{\boldsymbol{w}}_\ell^{\text{TF}}, \boldsymbol{x}_{test} \rangle$.**

Now we can make predictions on text query $\boldsymbol{x}_{\text{test}}$:

$$\begin{bmatrix} \boldsymbol{\xi} & \boldsymbol{x}_{\text{test}} \\ \hat{\boldsymbol{w}}_\ell^{\text{TF}} & \boldsymbol{x}_{\text{test}} \end{bmatrix} \xrightarrow{\text{mov}} \begin{bmatrix} \boldsymbol{\xi} & \boldsymbol{x}_{\text{test}} \\ \hat{\boldsymbol{w}}_\ell^{\text{TF}} & \boldsymbol{x}_{\text{test}} \\ \boldsymbol{0} & \hat{\boldsymbol{w}}_\ell^{\text{TF}} \end{bmatrix} \xrightarrow{\text{mul}} \begin{bmatrix} \boldsymbol{\xi} & \boldsymbol{x}_{\text{test}} \\ \hat{\boldsymbol{w}}_\ell^{\text{TF}} & \boldsymbol{x}_{\text{test}} \\ \boldsymbol{0} & \hat{\boldsymbol{w}}_\ell^{\text{TF}} \\ 0 & \langle \hat{\boldsymbol{w}}_\ell^{\text{TF}}, \boldsymbol{x}_{\text{test}} \rangle \end{bmatrix} \tag{35}$$

Finally, we can have an readout layer $\boldsymbol{\beta}_{\text{ReadOut}} = \{\boldsymbol{u}, v\}$ applied (see Definition 3.3) with $\boldsymbol{u} = [\boldsymbol{0}_{3d} \quad 1]^{\top}$ and $v = 0$ to extract the prediction $\langle \hat{\boldsymbol{w}}_\ell^{\text{TF}}, \boldsymbol{x}_{\text{test}} \rangle$ at the last location, given by $\boldsymbol{x}_{\text{test}}$. This is exactly how Iterative Newton makes predictions.

**To Perform $k$ steps of Newton's iterations, Transformers need $\mathcal{O}(k)$ layers.**

Let's count the layers:

- **Initialization**: mov needs $\mathcal{O}(1)$ layer; gathering $\alpha\boldsymbol{S}$ needs $\mathcal{O}(1)$ layer; and aff needs $\mathcal{O}(1)$ layer. In total, Transformers need $\mathcal{O}(1)$ layers for initialization.

- **Newton Iteration**: each exact Newton's iteration requires $\mathcal{O}(1)$ layer. Implementing $k$ iterations requires $\mathcal{O}(k)$ layers.

- **Implementing $\hat{\boldsymbol{w}}_\ell^{\text{TF}}$**: We need one operation of mov and mul each, requiring $\mathcal{O}(1)$ layer each. Apply Lemma B.3 for summation also requires $\mathcal{O}(1)$ layer.

- **Making prediction on test query**: We need one operation of mov and mul each, requiring $\mathcal{O}(1)$ layer each.

Hence, in total, Transformers can implement $k$-step Iterative Newton and make predictions accordingly using $\mathcal{O}(k)$ layers. $\qquad\square$

*Remark* B.5. We note that Giannou et al. [2023] used 13 layers to compute one Newton Iteration, and in our construction, we need only one Transformer layer (with one attention layer and one MLP layer) to compute one Newton Iteration. At the same time, we didn't use Akyürek et al. [2022] for constructing Newton Iterations. Akyürek et al. [2022] is applied to initialize Newton and for reading out the prediction.

In our construction, only the initialization and read-out prediction components use causal attention and softmax because Akyürek et al. [2022]'s construction is applied. To be more specific, those are the first 3 layers in initializing Iterative Newton and the last 5 layers in reading out the predictions from the computed pseudo-inverse. All the layers corresponding to the Iterative Newton updates are using full attention and normalized ReLU activations.

*Remark* B.6. We note that our proof can be extended to causal attention for $n$ sufficiently larger than $d$. Under causal attention (see Definition 3.1) with normalized ReLU activation, Equation (29) can be rewritten as follows, given $t > d$, we first choose $\boldsymbol{G} = \begin{bmatrix} \boldsymbol{O}_d & -\frac{1}{2}\boldsymbol{I}_d \\ \boldsymbol{O}_d & \boldsymbol{O}_d \end{bmatrix}$, where the coefficient on the upper right block is $-\frac{1}{2}$ instead of $-\frac{n}{2}$ originally. Then

$$
\begin{aligned}
\boldsymbol{h}_t^{(\ell+1)} &= \begin{bmatrix} \boldsymbol{T}^{(\ell)}\boldsymbol{x}_t \\ \boldsymbol{x}_t \end{bmatrix} + \frac{1}{t}\sum_{j=1}^{t} \begin{bmatrix} \boldsymbol{T}^{(\ell)}\boldsymbol{x}_j \\ \boldsymbol{0}_d \end{bmatrix} \begin{bmatrix} \boldsymbol{T}^{(\ell)}\boldsymbol{x}_j \\ \boldsymbol{0}_d \end{bmatrix}^\top \begin{bmatrix} -\frac{1}{2}\boldsymbol{x}_t \\ \boldsymbol{0}_d \end{bmatrix} \\
&= \begin{bmatrix} \boldsymbol{T}^{(\ell)}\boldsymbol{x}_t - \frac{1}{2}\frac{1}{t}\sum_{j=1}^{t}(\boldsymbol{T}^{(\ell)}\boldsymbol{x}_j)(\boldsymbol{T}^{(\ell)}\boldsymbol{x}_j)^\top\boldsymbol{x}_t \\ \boldsymbol{x}_t \end{bmatrix} \\
&= \begin{bmatrix} \boldsymbol{T}^{(\ell)}\boldsymbol{x}_t - \frac{1}{2}\boldsymbol{T}^{(\ell)}\left(\frac{1}{t}\sum_{j=1}^{t}\boldsymbol{x}_j\boldsymbol{x}_j^\top\right)\boldsymbol{T}^{(\ell)\top}\boldsymbol{x}_t \\ \boldsymbol{x}_t \end{bmatrix} \\
&= \begin{bmatrix} \left(\boldsymbol{T}^{(\ell)} - \frac{1}{2}\boldsymbol{T}^{(\ell)}\hat{\boldsymbol{\Sigma}}\boldsymbol{T}^{(\ell)\top}\right)\boldsymbol{x}_t \\ \boldsymbol{x}_t \end{bmatrix}
\end{aligned}
\tag{36}
$$

where $\hat{\boldsymbol{\Sigma}} = \frac{1}{t}\sum_{j=1}^{t}\boldsymbol{x}_j\boldsymbol{x}_j^\top$ is the estimate of the covariance matrix given seen in-context examples $\{\boldsymbol{x}_j, y_j\}_{j=1}^{t}$ so far. Since $t > d$, $\hat{\boldsymbol{\Sigma}}$ is an unbiased estimate for $\boldsymbol{\Sigma} \approx \frac{1}{n}\boldsymbol{S}$ if $n$ is sufficiently large. The rest of the proof follows similarly, up to the perturbation introduced by the error in the estimate of $\hat{\boldsymbol{\Sigma}}$.

We also note when $t < d$, the estimate $\hat{\boldsymbol{\Sigma}} = \frac{1}{t}\sum_{j=1}^{t}\boldsymbol{x}_j\boldsymbol{x}_j^\top$ is no longer a valid covariance matrix since it's singular. Then this gives different $\boldsymbol{T}^{(\ell+1)}$ for different time stamp $t < d$ and such error may propagate in our proof. Hence, a formal extension to causal models requires extensive analysis of the error bounds and it is beyond the scope of this work. Nonetheless, we provide a plausible direction of such an extension.

## B.3 Iterative Newton as a Sum of Moments Method

Recall that Iterative Newton's method finds $\boldsymbol{S}^\dagger$ as follows

$$
\boldsymbol{M}_0 = \underbrace{\frac{2}{\|\boldsymbol{S}\boldsymbol{S}^\top\|_2}}_{\alpha}\boldsymbol{S}^\top, \qquad \boldsymbol{M}_k = 2\boldsymbol{M}_{k-1} - \boldsymbol{M}_{k-1}\boldsymbol{S}\boldsymbol{M}_{k-1}, \forall k \geq 1.
\tag{37}
$$

We can expand the iterative equation to moments of $\boldsymbol{S}$ as follows.

$$
\boldsymbol{M}_1 = 2\boldsymbol{M}_0 - \boldsymbol{M}_0\boldsymbol{S}\boldsymbol{M}_0 = 2\alpha\boldsymbol{S}^\top - 4\alpha^2\boldsymbol{S}^\top\boldsymbol{S}\boldsymbol{S}^\top = 2\alpha\boldsymbol{S} - 4\alpha^2\boldsymbol{S}^3.
\tag{38}
$$

Let's do this one more time for $\boldsymbol{M}_2$.

$$
\begin{aligned}
\boldsymbol{M}_2 &= 2\boldsymbol{M}_1 - \boldsymbol{M}_1\boldsymbol{S}\boldsymbol{M}_1 = 2(2\alpha\boldsymbol{S} - 4\alpha^2\boldsymbol{S}^3) - (2\alpha\boldsymbol{S} - 4\alpha^2\boldsymbol{S}^3)\boldsymbol{S}(2\alpha\boldsymbol{S} - 4\alpha^2\boldsymbol{S}^3) \\
&= 4\alpha\boldsymbol{S} - 8\alpha^2\boldsymbol{S}^3 - 4\alpha^2\boldsymbol{S}^3 + 16\alpha^3\boldsymbol{S}^5 - 16\alpha^4\boldsymbol{S}^7 \\
&= 4\alpha\boldsymbol{S} - 12\alpha^2\boldsymbol{S}^3 + 16\alpha^3\boldsymbol{S}^5 - 16\alpha^4\boldsymbol{S}^7.
\end{aligned}
\tag{39}
$$

We can see that $\boldsymbol{M}_k$ are summations of moments of $\boldsymbol{S}$, with respect to some pre-defined coefficients from the Newton's algorithm. Hence Iterative Newton is a special of an algorithm which computes an approximation of the inverse using second-order moments of the matrix,

$$
\boldsymbol{M}_k = \sum_{s=1}^{2^{k+1}-1} \beta_s \boldsymbol{S}^s
\tag{40}
$$

with coefficients $\beta_s \in \mathbb{R}$.

We note that Transformer circuits can represent other sum of moments other than Newton's method. We can introduce different coefficients $\beta_i$ than in the proof of Theorem 5.1 by scaling the value matrices or through the MLP layers.

## B.4 Estimated weight vectors lie in the span of previous examples

What properties can we infer and verify for the weight vectors which arise from Newton's method? A straightforward one arises from interpreting any sum of moments method as a kernel method.

We can expand $S^s$ as follows

$$S^s = \left(\sum_{i=1}^{t} \boldsymbol{x}_i \boldsymbol{x}_i^\top\right)^s = \sum_{i=1}^{t} \left(\sum_{j_1,\cdots,j_{s-1}} \langle \boldsymbol{x}_i, \boldsymbol{x}_{j_1}\rangle \prod_{v=1}^{s-2} \langle \boldsymbol{x}_{j_v}, \boldsymbol{x}_{j_{v+1}}\rangle\right) \boldsymbol{x}_i \boldsymbol{x}_{j_{s-1}}^\top. \tag{41}$$

Then we have

$$
\begin{aligned}
\hat{\boldsymbol{w}}_t = M_t \boldsymbol{X}^\top \boldsymbol{y} &= \sum_{s=1}^{2^{t+1}-1} \beta_s \boldsymbol{S}^s \boldsymbol{X}^\top \boldsymbol{y} \\
&= \sum_{s=1}^{2^{t+1}-1} \beta_s \left\{\sum_{i=1}^{t}\left(\sum_{j_1,\cdots,j_{s-1}} \langle \boldsymbol{x}_i, \boldsymbol{x}_{j_1}\rangle \prod_{v=1}^{s-2} \langle \boldsymbol{x}_{j_v}, \boldsymbol{x}_{j_{v+1}}\rangle\right)\boldsymbol{x}_i \boldsymbol{x}_{j_{s-1}}^\top\right\}\left\{\sum_{i=1}^{t} y_i \boldsymbol{x}_i\right\} \\
&= \sum_{s=1}^{2^{t+1}-1} \beta_s \left(\sum_{i=1}^{t}\left(\sum_{j_1,\cdots,j_s} y_{j_s} \langle \boldsymbol{x}_i, \boldsymbol{x}_{j_1}\rangle \prod_{v=1}^{s-1} \langle \boldsymbol{x}_{j_v}, \boldsymbol{x}_{j_{v+1}}\rangle\right)\boldsymbol{x}_i\right) \\
&= \sum_{i=1}^{t} \underbrace{\left(\sum_{s=1}^{2^{t+1}-1} \sum_{j_1,\cdots,j_s} \beta_s y_{j_s} \langle \boldsymbol{x}_i, \boldsymbol{x}_{j_1}\rangle \prod_{v=1}^{s-1} \langle \boldsymbol{x}_{j_v}, \boldsymbol{x}_{j_{v+1}}\rangle\right)}_{\phi_t(i|\boldsymbol{X},\boldsymbol{y},\boldsymbol{\beta})} \boldsymbol{x}_i \\
&= \sum_{i=1}^{t} \phi_t(i \mid \boldsymbol{X}, \boldsymbol{y}, \boldsymbol{\beta})\, \boldsymbol{x}_i
\end{aligned}
\tag{42}
$$

where $\boldsymbol{X}$ is the data matrix, $\boldsymbol{\beta}$ are coefficients of moments given by the sum of moments method and $\phi_t(\cdot)$ is some function which assigns some weight to the $i$-th datapoint, based on all other datapoints. Therefore if the Transformer implements a sum of moments method (such as Newton's method), then its induced weight vector $\tilde{\boldsymbol{w}}_t(\text{Transformers} \mid \{\boldsymbol{x}_i, y_i\}_{i=1}^{t})$ after seeing in-context examples $\{\boldsymbol{x}_i, y_i\}_{i=1}^{t}$ should lie in the span of the examples $\{\boldsymbol{x}_i\}_{i=1}^{t}$:

$$\tilde{\boldsymbol{w}}_t(\text{Transformers} \mid \{\boldsymbol{x}_i, y_i\}_{i=1}^{t}) \stackrel{?}{=} \text{Span}\{\boldsymbol{x}_1, \cdots, \boldsymbol{x}_t\} = \sum_{t=1}^{t} a_i \boldsymbol{x}_i \qquad \text{for coefficients } a_i. \tag{43}$$

We test this hypothesis. Given a sequence of in-context examples $\{\boldsymbol{x}_i, y_i\}_{i=1}^{t}$, we fit coefficients $\{a_i\}_{i=1}^{t}$ in Equation (43) to minimize MSE loss:

$$\{\hat{a}_i\}_{i=1}^{t} = \underset{a_1, a_2, \cdots, a_t \in \mathbb{R}}{\arg\min} \left\|\tilde{\boldsymbol{w}}_t(\text{Transformers} \mid \{\boldsymbol{x}_i, y_i\}_{i=1}^{t}) - \sum_{t=1}^{t} a_i \boldsymbol{x}_i\right\|_2^2. \tag{44}$$

We then measure the quality of this fit across different number of in-context examples $t$, and visualize the residual error in Figure 27. We find that even when $t < d$, Transformers' induced weights still lie close to the span of the observed examples $\boldsymbol{x}_i$'s. This provides an additional validation of our proposed mechanism.

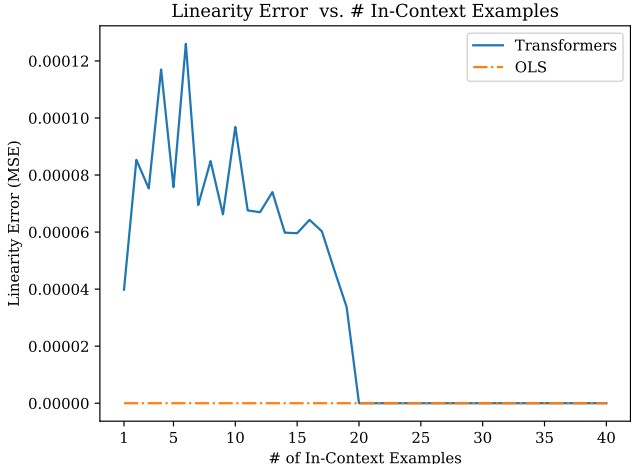

Figure 27: Verification of hypothesis that the Transformers induced weight vector $w$ lies in the span of observed examples $\{x_i\}$.

## C  Computes

All experiments involving fine-tuning GPT2 models to learn in-context linear regressions are trained on one NVIDIA A6000. Linear probing experiments also used one NVIDIA A6000.

## D  License

We used PyTorch Paszke et al. [2019] as our code framework and we used PyTorch implementation of LSTMs. PyTorch is licensed under the Modified BSD license.

We used GPT-2 Model as our backbone, and it's released under MIT License. We used trained GPT-2 checkpoints for linear regression by Garg et al. [2022] and it's released under MIT License.

## E  Limitations

In this work, our analyses of Transformers are mostly based on only one simple task: linear regression. It might not be able to extrapolate to any arbitrary algorithmic tasks. It would be interesting for future work to extend such analysis to an extensive class of problems.

## F  Broader Impacts

This paper presents work whose goal is to advance the field of Machine Learning. Through a mechanistic understanding of Transformers, the backbone of modern large language models (LLMs), this work can help advance building safe and trustworthy models.

