# OpenReview forum: "Transformers Learn to Achieve Second-Order Convergence Rates for In-Context Linear Regression"
_NeurIPS.cc/2024/Conference — NeurIPS 2024 poster_

### Official Review · Reviewer_J6bG · 2024-06-25

**Soundness:** 3
**Presentation:** 3
**Contribution:** 2
**Rating:** 3
**Confidence:** 3

**Summary:**

The authors show that when trained on ICL tasks encoding a regression problem, the layers of a transformer implement an implicit higher order optimization method. They contrast this with prior work on this subject that suggest that the transformer in this setting implicitly implements a preconditioned gradient method. They also construct a transformer capable of implementing this proposed algorithm.

**Strengths:**

There appears to be a better match between the layers of the transformer and iterations of iterative newtons method as compared to the iterations of the gradient methods they consider.

**Weaknesses:**

1. In the comparisons to Iterative Newton: If the Transformer had 40 layers do you expect that each layer would implement a single iteration or that it would match more layers of IN overall? In other words, is there something fundamental about taking 3 layers together to compute one iteration?

2. Does the transformer also match the higher order methods when solving other problems? That is, if the objective was not OLS for linear regression? The problem with studying only linear regression is that subsequent layers interpolating between 0 and the OLS solution is somewhat what one would “expect”. Not matching with GD fits into this interpretation because GD exhibits an oscillatory behavior for ill-conditioned problems that an “interpolator” would have no reason to match.
    1. What happens to Appendix A.3 when we compare the transformer to the appropriately preconditioned GD?
    2. Figure 15 is only an illustration of the best matching iterations. I have two concerns with this. The construction given in the paper is theoretical and not necessarily the one used by actual transformers (it is also deeper than 3 layers). I think this reduces the impact of saying that iterative newton matches every third layer of the transformer, since even GD is matching (though with a varying number of iterations). And the match quality of GD after those varying number of iterations is not much worse.

**Questions:**

1. What happens to the covariates across layers? To implement these algorithms, the transformer needs to implicitly compute some second order moment of the original data, but at each layer can only work with the “modified covariates” at that layer. It would be interesting if these quantities somehow turned out to be invariant across layers.
2. Is it fair to say that ReLU activations don’t hurt performance? Softmax seems to be pretty crucial, or why would people go to the trouble of implementing something that has a larger run-time? Even most of the works cited only say it isn’t much worse, sometimes. Also, it seems softmax is used for the experiments in this work anyway, and ReLU is only used for some of the layers in the construction. imo this is not a big issue, just pointing it out.
3. I dont understand the description of the Iterative Newtons Method (Page 3). It seems $S$ is an empirical covariance, and then $M_k$ is just a function of $S$. What is the purpose of computing the intermediate $\hat{w}_k^{\text{Newton}}$ (other than to show that they match the Transformer)?

**Limitations:**

Yes

---

> ### Author Rebuttal · Authors · 2024-08-06
>
> We thank the reviewer for detailed comments and suggestions. We are pleased that the reviewer agrees that our experiments show a better match between the layers of the transformer and iterations of iterative newtons than when compared with gradient descents.
>
> ## **Matching between Transformer layers and Iterative Newton steps**
> We believe this is some degree of misunderstanding by the reviewer. As shown in the heatmap in Figure 3 (left), one Transformer layer is implementing 3 iterations of Newton steps, not the other way around as the reviewer suggested.  The reviewer can also take a look at Figure 21 on a 24-layer transformer model, where each layer is approximately implementing one Newton step until convergence.
>
> ## **Generalization to other problems**
> We believe that linear regression is the right starting point to understand how transformers perform in-conext learning. It would be interesting to see the extensions to logistic regression, classification, and even non-convex problems. However, this is beyond the scope of this paper.
>
> ## **Not matching GD is expected**
> Garg et. al. [1] empirically showed that Transformers, when performing in-context linear regression, match the ordinary least-squares (OLS) solution. However, what remains unclear and what is the main focus of this paper, is **how** Transformers converge to the OLS solution.
> We wouldn’t say not matching GD is expected because many existing work claims that Transformers converge to the OLS solution via Gradient Descent (GD) [2,3,4,5]. The main contribution of this work is to show that (1) empirically, Transformers can converge exponentially faster than GD, which suggests that they emulate higher-order optimization methods; (2) theoretically, there is a construction of Transformers to implement a particular higher-order optimization method – the Iterative Newton’s method.
>
> ## **Comparison with Precondition GD**
> We would like to emphasize that even with well-conditioned data, our experiments show Transformers and Newton converge exponentially faster than Gradient Descent. Under the well-conditioned case, preconditioning would not do anything because the preconditioner (inverse of the Hessian matrix) is identity. Even under the ill-conditioned case, where the preconditioner is not identity, one needs to compute the inverse of the Hessian matrix first, where such inverse computation is already computationally heavy and involves second-order information, and to compute the inverse efficiently, one needs to use the Iterative Newton’s Method. In our setup, the eigenbasis of the covariance matrix $\Sigma$ is sampled at random, so there is no way the Transformer stores the preconditioner during training so they need to compute the inverse at ICL inference time.
>
> ## **Weakness 2.2: mismatch between theoretical construction and empirical Transformers**
> There is some degree of misunderstanding from the reviewer about our main claim and we would like to reiterate our claim and evidence.
>
> - First, we don’t claim that Iterative Newton is the algorithm that a trained Transformer model is implementing. What we really claim is that Transformers learn some algorithm that convergence exponentially faster than the gradient descent method, and algorithms with such log log convergence rate fall into the same category of “higher-order” optimization methods, where Iterative Newton is one of them. We do believe that convergence rate is a solid categorization of optimization algorithms. We also note the $\Omega(\log(1/\epsilon))$ lower bound of gradient-based methods [1] and it is not possible to improve Gradient Descent’s convergence rate *without using second-order methods*.
>
> - Second, our empirical results show that Transformers share the same convergence rate as Iterative Newton and such results can categorize Transformers as a higher-order method, and they are all exponentially faster than Gradient Descent algorithms. Please refer to Fig 3 for the well-conditioned case, Fig 15 for the ill-conditioned case, and Fig 21 for deeper transformers. The intuition why we don’t believe Transformers are doing GD is that, for example in Fig 3 (right), there is no way for one Transformer layer (from layer 7 to layer 8) to implement 500 gradient descent steps.
>
> - Finally, we show expressivity results that Transformers can indeed implement Iterative Newton’s method, a representative higher-order method.
>
> ## **Covariates across layers**
> We would like to humbly ask the reviewer to clarify the question. We also would like to point to our theoretical proof that the intermediates $M_k$ do not need to be stored in each layer. In this case, it will be difficult to extract the exact intermediate covariates.
>
> ## **ReLU activation**
> As the reviewer pointed out, this is not a bit issue as many existing work studies either linear attention (no activation applied to attention layers) or with ReLU activations.
>
> ## **Purpose of computing $w^{\mathrm{Newton}}_{k}$**
> It allows us to compute the convergence rate of algorithms, for which one needs to show the relationship between $||w_{k+1} - w_{\star}||$ and $||w_{k} - w_{\star}||$ for the optimal solution $w_{\star}$. In this case, we need to compute the intermediate
> $w^{\mathrm{Newton}}_{k}$.
>
> Moreover, $w^{\mathrm{Newton}}_{k}$ is needed to make predictions on any test sample, and we can measure how good each iteration has been by looking at the errors on test samples.

---

> > ### Comment · Reviewer_J6bG · 2024-08-08
> >
> > Thank you for your response.
> >
> > I did have a misunderstanding that several layers in the transformer are used to implement a single iteration of the higher order method. Rather it is the other way. I was coming at this from the perspective that perhaps there are some weights that implement the algorithm you propose, (like Theorem 5.1, but one the transformer actually learns).
> >
> > My main concern is that the observations here *might* be very specific to OLS for linear problems. The way it is solved traditionally (without transformers) this is just a quadratic problem. There have been many works in this space (those cited in the section on "**Do Transformers implement Gradient Descent?**"), but as far as I know, none of them consider learning any other function class, whereas this seems like a very natural question. Since a lot of these observations are empirical, it would make the message more powerful if these results could be demonstrated on non-linear regression problems.
> >
> > Considering the prior work, I think this paper presents an interesting ``closer look". But I wonder if we can hope that these insights would extend even to problems that are not quadratic. Would it be feasible in the remaining time to get a heat map for the similarity in weight when the ground truth is a single ReLU neuron, or the similarity in errors when the ground truth is a small MLP?
> >
> > Regarding the question about the covariates, what I was getting at is that an $L$ layer transformer can be thought of as an $L-1$ layer transformer with inputs that are the outputs of the actual first layer. I was just wondering if the new $x_i, y_i$ can be directly interpreted in any way. There must be some way that you can view the last $L-1$ layers as running an optimization algorithm to solve the problem presented by the outputs of the first layer. Is this optimization problem the same as the one the full transformer is running (but with one (or three) less iterations)?

---

> ### Author Response · Authors · 2024-08-06
> **References**
>
> [1]Shivam Garg, Dimitris Tsipras, Percy Liang, Gregory Valiant. *What Can Transformers Learn In-Context? A Case Study of Simple Function Classes*. In NeurIPS, 2022
>
> [2] Johannes von Oswald, Eyvind Niklasson, E. Randazzo, Joao Sacramento, Alexander Mordvintsev, Andrey Zhmoginov, and Max Vladymyrov. *Transformers learn in-context by gradient descent.* In International Conference on Machine Learning, 2022.
>
> [3] Ekin Akyurek, Dale Schuurmans, Jacob Andreas, Tengyu ¨ Ma, and Denny Zhou. *What learning algorithm is incontext learning? investigations with linear models.* The Eleventh International Conference on Learning Representations, 2024
>
> [4] Kwangjun Ahn, Xiang Cheng, Hadi Daneshmand, and Suvrit Sra. *Transformers learn to implement preconditioned gradient descent*. Advances in Neural Information Processing Systems 36. 2024
>
> [5] Yu Bai, Fan Chen, Haiquan Wang, Caiming Xiong, and Song Mei. *Transformers as statisticians: Provable in-context learning with in-context algorithm selection.* Advances in neural information processing systems, 36. 2024

---

> ### Author Response · Authors · 2024-08-11
> **Additional Non-Linear Experiments**
>
> Thank you for your thoughtful suggestions and comments.
>
> We have conducted an additional experiment on a non-linear function class: 2-layer neural network with ReLU activations, a setting also studied by pioneering work, Garg et. al. (2022) in Fig. 5(c) in their NeurIPS 2022 paper version. Please refer to https://anonymous.4open.science/api/repo/transformer_icl_neurips_rebuttal-2E55/file/Transformer_ICL_rebuttal_additional.pdf for experimental results. **TL;DR**: Transformers still converge superlinearly, and exponentially faster than GD -- indicating higher-order optimization in-context.
>
> In the same prompt setup $(x_1, y_1, \cdots, x_t, y_t)$, each $y_k$ is generated by $y_k = a^\top \mathrm{ReLU}(Wx_k + v) + b$. There are 100 neurons, aka, the hidden size, and the problem dimension $d$ is still 20. As shown in Figure 3a, even on 2-layer neural network tasks, Transformer shows superlinear convergence rates. As shown in Figure 3b, Transformer shows an exponentially faster convergence rate than GD’s, because GD’s steps are shown in log scale and the trend is linear – similar to Figure 9 in the main paper.
>
> Although our title indicates the study mainly focuses on linear functions, we find the extension to non-linear functions quite meaningful. We will make the experiments extending to non-linear function classes more thorough and include more discussions in the camera-ready version.

---

> ### Author Response · Authors · 2024-08-11
> **Question about the covariates**
>
> In analogy to many optimization algorithms, what's updated along the iterations are the estimators $\hat{w}$ than the covariates $x_i, y_i$.  We can understand by analogy to any other iterative method, such as GD. The final $L-1$ steps of GD solve some problem whose input is the solution to the first step, $\hat{w}_1$. However, the $x_i, y_i$ for the linear regression instance would not change, it’s just that the optimization algorithm starts with a better estimate. In analogy to Newton, the goal is to compute better and better estimates of the inverse covariance iteratively. Please let us know if you have further questions.

---

> ### Author Response · Authors · 2024-08-13
>
> Dear Reviewer, as the discussion period is about to end, we would like to know if our newest experiments on two-layer MLP with ReLU activation (thread here: https://openreview.net/forum?id=L8h6cozcbn&noteId=7Gu8rWpfSn) have cleared out any of your concerns. We are also happy to answer any further questions. Thanks!

---

### Official Review · Reviewer_p5Dq · 2024-07-09

**Soundness:** 3
**Presentation:** 4
**Contribution:** 3
**Rating:** 7
**Confidence:** 4

**Summary:**

This paper demonstrates that transformers learn to approximately perform higher order algorithms, building on the previous works that demonstrate that transformers can approximate gradient descent which is a first-order algorithm. For the problem of linear-regression, it is empirically shown that that prediction errors of transformer are similar to that of iterative Newton's method, and so are their corresponding rates of convergence. Further, it is shown by explicit construction of weights how a $k+\mathcal{O}(1)$ layers of transformers can implement $k$ steps of the Newton's method.

**Strengths:**

(1) The work provides a very good understanding of how transformers can efficiently solve (even ill-conditioned) linear regression problems, which is of interest to the research community.

(2) This paper establishes that transformers are better algorithmic engines than other deep learning models (for example, LSTMs, in Figure 6 of Appendix A.2.1).

(3) The performance of the trained transformer is very close to that of a higher-order algorithm, even though transformer is not trained for different number of layers $\ell$ separately (only the ReadOut layer is retrained). This demonstration strongly supports the crucial claim that the representations after each layers learned by the transformers in fact mimic the updates of some higher-order algorithm.

(4) The experiments are well-executed and their presentation is commendable.

(5) The proof of the main theoretical contribution (Theorem 5.1) is very clean.

**Weaknesses:**

(1) Although the result is very interesting that leads to a better understanding of the abilities of a transformer, there is little direct impact in terms of application.

**Questions:**

(1) The effect of the number of layers is that the prediction after each layer progressively improves. However, one might also be interested in the effect of embedding dimension on the rate of convergence, which is not discussed. I am curious if you have any comments on this?

**Limitations:**

(1) Within the scope of the problem, there is no specific limitation that requires attention (pun intended).

---

> ### Author Rebuttal · Authors · 2024-08-06
>
> We thank the reviewer for their detailed comments and suggestions. We are pleased they find our work valuable in demonstrating how transformers efficiently solve linear regression problems, including ill-conditioned ones. They appreciate our comparison showing transformers as superior to models like LSTMs, and acknowledge our experimental results show strong evidence that transformers resemble higher-order algorithms. We thank the reviewer for commending our well-executed experiments, clear presentation, and clean proof of our main theorem.
>
>
> ## **Direct Impact**
> We acknowledge that this is a more interpretability and theoretical work whose direct impact to applications is unknown. However, we believe that interpretability could be useful in general, for example, it facilitates our understanding of architectures, and motivates us to make reliable changes in the future.
>
> ## **Impact of Hidden Dimension Size**
> Empirically, we ablate 12-layer 1-head transformers with different hidden dimensions: 64, 32, 16, and 8. Please see results in the Figure 2 in the rebuttal PDF. We find that transformers with hidden dimensions 64 and 32 are able to converge to the OLS solution. On the contrary, transformers with hidden dimensions 16, and 8 could not.
> Theoretically, as stated in Theorem 5.1, the hidden dimensions in our theoretical construction require a size of $\mathcal O(d)$, where $d$ is the dimension of the linear regression problem. That being said, as long as the hidden dimensions are of order $\mathcal O(d)$, transformers are able to converge to the OLS solution, and the higher-order optimization methods are learned in-context. This coincides with our empirical results, where $d = 20$ and the hidden dimensions need to be $\mathcal O(d)$.

---

> > ### Comment · Reviewer_p5Dq · 2024-08-12
> >
> > I read the rebuttal and my doubts have been addressed adequately. I shall maintain my score. Thank you.

---

> > > ### Author Response · Authors · 2024-08-13
> > >
> > > Dear reviewer, Thank you for discussing with us and we appreciate your support for our paper.

---

### Official Review · Reviewer_dusD · 2024-07-11

**Soundness:** 3
**Presentation:** 3
**Contribution:** 3
**Rating:** 7
**Confidence:** 3

**Summary:**

This work investigated the ability of transformers to implement higher-order optimization methods for in-context learning of linear regression tasks. The authors considered a noiseless linear regression setting and compared the output of each layer of TF with few steps on gradient descent (GD), online gradient descent (OGD), iterative Newton's method, and empirically showed that the output has a linear trend with the iterations of the Newton's method but an exponential trend with GD's iterations. Therefore, they concluded that TFs are more likely to implement high-order methods instead of first-order method for ICL tasks. They also performed experiments on ill-conditioned tasks and observed similar trends. In addition to the experiments, they  theoretically proved that there exists a polynomial size TF that can implement  iterative Newton's method.

**Strengths:**

The paper is well-written. The experimental results and their implications are well-discussed. The similarity metrics used to compare different algorithms are reasonable. Novel theoretical results on the approximation ability of TFs are proved.

**Weaknesses:**

Assumption on the linear regression tasks. This work considered a noiseless setting in which the optimal predictor in the OLS. It is unclear to what extent the observations in this work can be generalized the setting where the noise exists.

**Questions:**

What is the choice of stepsize for GD? Is it optimal? Would different choices of stepsize for GD affect the exponential trend  shown in Figure 3 (right)?

Do similar observations appear in linear regression tasks with non-zero noise? For example, in this case, would TF perform high-order optimization methods to learn the optimal ridge predictor?

**Limitations:**

This work compared the outputs of TFs and some first-order/higher-order algorithms and concluded that TFs are more likely performing higher-order optimization based on the similarity of their outputs. However, this does not imply that TFs  internally implement Newton's method or any other specific higher-order method.

This work focused on ICL with linear regression tasks. It is unclear if TFs still behave like higher-order methods in other convex ICL tasks (e.g., logistic regression for classification) and what TFs approximate if the learning task is non-convex.

---

> ### Author Rebuttal · Authors · 2024-08-06
>
> We thank the reviewer for detailed comments and suggestions. We are pleased that the reviewer thinks our experimental results and their implications are well-discussed. We are happy to see that the reviewer regards our theoretical results novel, our comparison metrics reasonable, and our paper well-written.
>
> ## **Generalization to noisy problems**
> We thank the reviewer for this suggestion. We run experiments on noisy linear regression with a noise standard deviation of $\sigma = 0.1$. Please refer to Figure 1 in the rebuttal PDF for detailed results. We observe that in noisy setup, transformers and iterative Newton’s method still share the same convergence rate, and are both exponentially faster than GD.
>
> Additionally, under the noisy linear regression setup, the closed form solution would be $\hat{w} = (X^\top X + \lambda I)^\dagger X^\top y$. Our theoretical construction stills holds with slight modifications: replacing $S = X^\top X$ by $S = X^\top X + \lambda I$. We will discuss this more clearly in our next revision.
>
>
> ## **Choice of step size for GD**
> Notice that the optimal step size for GD is $\eta = \frac{2}{\beta + \gamma}$ where $\beta$ and $\gamma$ are the smallest and largest eigenvalues for $S = X^\top X$ respectively. However, each sequence is sampled randomly and will have different optimal $\eta$, and it would be an unfair comparison if extra computations are allowed for eigenvalues of $S$ – which involves computing higher moments of $S$ if using numerical methods, for example, power methods. To circumvent this, we sampled 10,000 sequences and computed the average optimal step size $\bar{\eta} = \mathbb E[\eta]$ and used the same step size $\bar{\eta}$ for GD.
>
> Empirical, we showed that transformers match Iterative Newton’s convergence rate. Theoretically, Iterative Newton’s convergence rate is exponentially faster than GD with optimal step sizes (see lines 121 and 130). Combining the two, we can deduce that transformers will also be exponentially faster than GD with optimal step sizes.
>
>
> ## **Generalization to logistic regression and classification, or even non-convex problems**
> We believe that linear regression is the right starting point to understand how transformers perform in-conext learning. It would be interesting to see the extensions to logistic regression, classification, and even non-convex problems. However, this is beyond the scope of this paper.

---

> > ### Comment · Reviewer_dusD · 2024-08-13
> >
> > Thanks for the clarification and additional experiments. My questions have been answered and I will update my score.

---

> > ### Author Response · Authors · 2024-08-13
> > **Thanks**
> >
> > Dear reviewer, Thank you for discussing with us. We appreciate your updated score and your support for our paper.

---

### Official Review · Reviewer_F4MD · 2024-07-16

**Soundness:** 3
**Presentation:** 3
**Contribution:** 3
**Rating:** 6
**Confidence:** 3

**Summary:**

In the paper the authors study the nature of in-context learning in transformers. Starting from the hypothesis of previous work on the fact that transformers may internally implement gradient descent algorithms to correctly perform linear regression on test data, the authors put forward the theory that transformers actually implement a different, higher-order algorithm. The authors back their claim with empirical evidence on the convergence rate of the algorithm across layers, and they provide a theoretical construction for the implementation of Newton's iterative algorithm using the transformer architecture.

**Strengths:**

I believe that the idea that transformers implement a higher-order algorithm for linear regression is intriguing, and I think that the empirical evidence provided by the authors definitely rule out the possibility that transformers simply implement gradient descent and not a more sophisticated algorithm.

**Weaknesses:**

The limit of the work is two-fold: (1) the authors only provide circumstantial -- and not direct -- evidence that the network is learning a higher-order algorithm, and the provided evidence on ill-conditioned data actually seem to contradict the theory that the network is actually implementing Newton's iterative method and not something ever more sophisticated; (2) there is quite a bit of mismatch between the empirical and the theoretical results: there is a 8-layer baseline required for the proposed construction to work that doesn't seem to appear in the experiments, and 8 is quite a large number for the setting presented in the paper (e.g., in Figure 2 the transformer seem to achieve small error already after 8 layers).

**Questions:**

What do you think may be causing the mismatch between the theoretical and the empirical results, i.e., what do you think is the limit of your construction? Do you think that it may be improved, e.g, by some additional architecture components, or by the substitution of softmax instead of ReLU?

---

> ### Author Rebuttal · Authors · 2024-08-07
>
> We thank the reviewer for detailed comments and suggestions. We are pleased that the reviewer thinks our ideas intriguing and empirical evidence convincing.
>
> ## **Evidence for higher-order is indirect**
> Our main evidence is the convergence rate. We do believe that convergence rate is a solid categorization of optimization algorithms where higher-order methods will have $\log\log(1/\epsilon)$ rates whereas first-order could only achieve $\log(1/\epsilon)$ rates. We also note the $\Omega(\log(1/\epsilon))$ lower bound of gradient-based methods and it is not possible to improve Gradient Descent’s convergence rate *without using second-order methods*.
>
> ## **contradiction in ill-conditioned case**
> There might be some degree of misunderstanding about our ill-conditioned experiments. Our experiments show that both Transformers and Newton’s method are *not* susceptible to ill-conditioned data – a common trait for higher-order optimization methods. In both well-conditioned and ill-conditioned experiments, our heatmaps (Fig 3 and Fig 15) show that one Transformer layer could implement three Newton iterations, which indicates that Transformers could implement some algorithms even more complicated and stronger than Iteratvie Newton’s method. However, they are still higher-order optimizations with a similar convergence rate to Newton’s method,
>
> ## **Mismatch between the empirical and the theoretical results**
> Surely, there’s a mismatch between empirical and theoretical results for the initial constant number of layers needed, but they are both  $\mathcal O(1)$. Empirically, we observe there’s an  $\mathcal O(1)$ number of layers needed for a **warmup**: the initial layers where transformers cannot show improvements in prediction errors. Please see Figure 2(a) for 12-layer transformers, 2 layers at the beginning are reserved for warmup; in Figure 21 for 24-layer transformers with iterative newton, 4 layers are reserved at the beginning. Similarly, our theoretical construction needs 8 layers, but it still lands in the category of $\mathcal O(1)$.
>
> Our main focus for theoretical construction is to show that, to implement $k$ Newton steps, we need $k + \mathcal O(1)$ Transformers layers, with a particular focus on the 1-to-1 correspondence between **intermediate layers** and Newton steps. Admittedly, we didn’t optimize our construction for the $\mathcal O(1)$ constant number of layers needed to initial layers. There could be alternative constructions with a smaller amount of warmup layers than 8, but that’s not the main focus of our theoretical analysis.
>
> There could be many causes for the mismatch of the constant. For example, the number of layers of the Transformers, the hidden dimension size, etc. At the same time, optimization algorithms used to train these Transformers could also be another factor on the empirical side. On the theoretical side, as the reviewer pointed out, the substitution of softmax activation by ReLU could be an important factor. Nonetheless, matching the exact number between empirical and theoretical results would be tedious but matching the number within the same complexity of  $\mathcal O(1)$ is simply what’s presented in this paper.

---

> > ### Comment · Reviewer_F4MD · 2024-08-09
> >
> > Thank you for your valuable comments. I will keep my positive score.

---

> > > ### Author Response · Authors · 2024-08-13
> > >
> > > Dear reviewer, Thank you for discussing with us and we appreciate your support for our paper.

---

### Official Review · Reviewer_vdXr · 2024-07-22

**Soundness:** 4
**Presentation:** 4
**Contribution:** 2
**Rating:** 7
**Confidence:** 4

**Summary:**

This paper studies in-context learning in transformers, and find that gradient descent converges more slowly than transformers/Newton's method for linear regression. Previous works proposed that transformers do in-context learning using gradient-based algorithms, based on experiments with linear regression tasks. This paper suggests that transformers use a higher-order optimization method rather than a first-order method, for linear regression. The evidence for this is as follows: (1) transformers converge to the OLS solution at a similar rate as iterative newton. The predictions after each layer match those of Newton's method after a proportional number of iterations. (2) GD requires a much larger number of steps (exponentially many) to match the transformer's errors.

The problem setup/experimental setup is similar to prior works in this area - in each example sequence, they sample a weight vector $w$, and a matrix $\Sigma$ which is from some distribution over PSD matrices (in some of the experiments, $\Sigma$ is an identity matrix, and in others, $\Sigma$ is ill-conditioned). The examples $x_i$ within the sequence are sampled from a Gaussian with covariance $\Sigma$. They use the following metrics to measure similarity between transformers/other algorithms (namely Newton's method and GD):
1. Similarity of errors - cosine similarity of the errors that two algorithms achieve. On a given data sequence of length $n$, they take the vector of errors that each algorithm achieves on $y_2, \ldots, y_{n + 1}$, where for $y_i$, the algorithm is given $x_1, y_1, ..., x_i$. Then, they take the cosine similarity between these two error vectors.
2. Similarity of induced weights - fit a weight vector to the predictions of this model, and compare two algorithms by comparing their induced weight vectors (on the same sequence of in-context examples).
3. Matching steps between algorithms - Let $p_a$ be a particular number of steps taken by algorithm A. Then, the best matching number of steps for algorithm B is the one that maximizes the cosine similarity with algorithm A at step $p_a$.

The results in Figure 2 show that the middle layers of transformers converge at a superlinear rate, similarly to Newton's method. Figure 2 also shows that gradient descent has a sub-exponential convergence rate, and gets slower as the number of steps increases. It requires a much larger number of iterations. They also match the steps between transformers and Newton's method and GD (in other words, for each layer L of the transformer, find the step of Newton's method/GD whose predictions are the most similar to the transformer at layer L). Figure 3a shows that there is a linear relationship between the transformer layer index and the corresponding step index of Newton's method. (Figure 8, which matches steps based on weight vectors instead of errors, seems to show a stronger linear trend.)

**Strengths:**

- The experiments are thorough and successfully show that transformers can learn optimization algorithms much faster than gradient descent.
- The paper is very well-written.

**Weaknesses:**

- As the authors mention, the experiments ultimately show that transformers behave more similarly to some higher-order method, rather than Newton's method in particular. In some cases, transformers can behave differently from Newton's method as well.
    - For instance, in Figure 20 (left), it could be argued that the transformer converges even faster than iterative Newton's method, i.e. the relationship between the iteration of Newton's method and the transformer layer index is not exactly linear. For instance, layer 14 of the transformer seems to correspond to about 4 iterations of Newton's method.

- It would be useful to clarify the difference between the takeaway of this paper and e.g. Garg, et al. (2022), which finds that transformers can match ordinary least-squares very closely.
    - If this submission shows that transformers learn Newton's method in particular, then the difference would be clear. This work gives evidence that transformers learn faster than GD, but somewhat less strong evidence that transformers learn a particular algorithm.

Garg, et al. (2024) "What Can Transformers Learn In-Context? A Case Study of Simple Function Classes"

==================================

The authors clarified my questions in the rebuttal - I am increasing my score from 6 to 7.

**Questions:**

- Figure 5, left, seems to be showing a drastic difference between the performance of transformers and Newton's method. This may be because the red curve only uses 5 steps of Newton's method. Is the result different if more steps of Newton's method are used?

**Limitations:**

Yes.

---

> ### Author Rebuttal · Authors · 2024-08-06
>
> We thank the reviewer for detailed comments and suggestions. We are pleased that the reviewer thinks our experiments are thorough and successfully support our main claim. We are also happy to see the reviewer thinks our paper is well-written.
>
> ## **In some cases, Transformers can behave differently from Newton's method**
> Admittedly, Transformers could behave slightly differently compared to Newton’s method as the reviewer pointed out. This could come from optimization noises during training or probing. Additionally, we are not claiming trained Transformers models are exactly implementing Iterative Newton’s method and this is why our title says “higher-order method” rather than “Newton’s Method”. Our main claim is that Transformers resembles a class of optimization algorithms, and a slight deviation from Newton’s method wouldn’t hurt such claims.
>
> ## **Clarify the difference between this paper and Garg et. al.**
> Garg et. al. empirically showed that Transformers, when performing in-context linear regression, match the ordinary least-squares (OLS) solution. However, what remains unclear is **how** Transformers converge to the OLS solution. One common hypothesis is that Transformers converge to the OLS solution via Gradient Descent (GD), while the main contribution of this work is to show that (1) empirically, Transformers can converge exponentially faster than GD, which suggests that they emulate higher-order optimization methods; (2) theoretically, there is a construction of Transformers to implement a particular higher-order optimization method – the Iterative Newton’s method.
>
> ## **Whether Transformers evidently learn a particular algorithm**
> Again, we are not claiming trained Transformers models are exactly implementing Iterative Newton’s method. We also compared Transformers with many alternative higher-order optimization methods such as Conjugate Gradient and (L-)BFGS, and the conclusion is that Transformers are learning a particular class of algorithms, that are utilizing higher-order information.
>
> ## **LSTM only matches 5-step Newton’s Method**
> Yes. As shown in Fig. 6(c), LSTMs couldn’t improve with more layers and the errors are still quite large. This implies that LSTMs could only converge to an estimator $\hat{w}_{LSTM}$ that achieves the same performance as Iterative Newton’s method after only 5 steps, which is far from converging to the least squares solution.

---

> > ### Comment · Reviewer_vdXr · 2024-08-12
> >
> > Thank you for the reply. I agree that this work goes beyond Garg, et al. by explicitly analyzing how the rate at which transformers converge to the OLS solution, depends on the number of layers. I will update my score.

---

> > > ### Author Response · Authors · 2024-08-13
> > > **Thanks!**
> > >
> > > Dear reviewer, Thank you for discussing with us. We appreciate your updated score and your support for our paper.

---

### Author Rebuttal · Authors · 2024-08-06

We sincerely thank the reviewers for their time and effort in reviewing our work. We are encouraged that the reviewers praise our work for providing a very good understanding of how transformers can efficiently solve linear regression (p5Dq) and find our ideas of higher-order optimization intriguing (F4MD). We are happy to see that reviewers find our experiments thorough (vdXr) and well-executed (p5Dq), our results reasonable (J6bG) and convincing (vdXr, F4MD), and their implications are well-discussed (dusD). We are also pleased to see the reviewer commend our theoretical results (dusD, p5Dq). Finally, we are delighted to see reviewers find our paper well-written (vdXr, dusD) and our proof clean (p5Dq).

We conducted two additional experiments per the reviewers’ comments and the results are attached in the **PDF**:

- **Noisy Linear Regression**. As reviewer dusD suggested, we conduct additional experiments on noisy linear regression problems, and find our claim that transformers learn high-order optimization methods in-context still holds in noisy linear regression setups.

- **Varying Hidden Dimension Sizes**. As reviewer p5Dq suggested, we ablate on the hidden dimensions of transformers, and observe the resonance between empirical results and our theoretical claim that a hidden dimension of $\mathcal O(d)$ is necessary for Transformers to converge to the OLS solution and thus learn higher-order optimization methods in-context.

We have addressed the comments in the responses to each reviewer and will incorporate all feedback into the paper.

---

### Decision · Program_Chairs · 2024-09-25

**Decision:**

Accept (poster)

**Comment:**

The paper analyzes in-context learning of linear models, showing that parametric transformers can achieve a super-linear convergence rate. To explain this rapid convergence, this paper demonstrates that transformers are expressive enough to implement the "iterative Newton's method" which enjoys a super-linear convergence rate.

Based on the discussions and reviews, I recommend to accept a poster presentation with the following revisions in the final draft:

- Narrowing the focus in the title, abstract, and introduction to linear regression. The implementation of second-order optimization for in-context learning, as claimed in the abstract, is beyond the paper's scope.
- Discussing the gap between theoretical analyses and experiments, including the differences between the architectures used in theory and those used in experiments.
- Including results for linear attention mechanisms, and then extending the analysis to attention layers with ReLU.